



# Multi-layer coupling between SURFEX-TEB-V9.0 and Meso-NH-v5.3 for modelling the urban climate of high-rise cities

Robert Schoetter[1], Yu Ting Kwok[2], Cécile de Munck[1], Kevin Ka Lun Lau[3], Wai Kin Wong[4], and Valéry Masson[1]

[1]CNRM, Université de Toulouse, Météo-France, CNRS, 42 avenue Gaspard Coriolis, 31057, Cedex 1, Toulouse, France
[2]School of Architecture, The Chinese University of Hong Kong, Hong Kong, China
[3]Institute of Future Cities, The Chinese University of Hong Kong, Hong Kong, China
[4]Hong Kong Observatory, Hong Kong, China

**Correspondence:** Robert Schoetter (robert.schoetter@meteo.fr)

**Abstract.** Urban Canopy Models (UCMs) represent the exchange of momentum, heat, and moisture between cities and the atmosphere. Single-layer UCMs interact with the lowest atmospheric model level and are suited for low- to mid-rise cities whereas multi-layer UCMs interact with multiple levels and can also be employed for high-rise cities. The present study describes the multi-layer coupling between the UCM Town Energy Balance (TEB) included in the land surface model SURFEX

and the mesoscale atmospheric model Meso-NH. This is a step towards better high-resolution weather prediction for urban areas in the future and studies quantifying the impact of climate change adaptation measures in high-rise cities. The effect of the buildings on the wind is considered using a drag force and a production term in the prognostic equation for turbulent kinetic energy. The heat and moisture fluxes from the walls and the roofs to the atmosphere are released at the model levels intersecting these urban facets. No variety of building height at grid point scale is considered to remain the consistency between

the modification of the Meso-NH equations and the geometric assumptions of TEB. The multi-layer coupling is evaluated for the heterogeneous high-rise high-density city of Hong Kong. It leads to a strong improvement of model results for near-surface air temperature and relative humidity, which is due to better consideration of the process of horizontal advection in the urban canopy layer. For wind speed, model results are improved on average by the multi-layer coupling, but not for all stations. Future developments of the multi-layer SURFEX-TEB will focus on improving the calculation of radiative exchanges, which

will allow a variety of building heights at grid point scale to be taken into account.

## 1 Introduction

### 1.1 Background

Atmospheric models need to account for the influence of surfaces with very different physical characteristics like forests, deserts, oceans, glaciers, or urban areas on the atmosphere. Land Surface Models (LSMs) have been developed (Koster et al.

2006, Guo et al. 2006) to calculate the surface fluxes of momentum, energy, water, and substances based on the prognostic variables of the atmospheric models and the physical, chemical, or biological processes relevant for the surface-atmosphere





exchanges (Best et al., 2004). LSMs are frequently subdivided into tiles to better represent the variety of surface types at grid point scale (Giorgi and Avissar, 1997). The prognostic surface equations are solved separately for each tile and the fluxes towards the atmospheric model are aggregated. Examples of such LSMs are the Noah LSM (Chen and Dudhia, 2001), the

Community Land Model (Oleson et al., 2010), and the Externalised Surface (SURFEX, Masson et al. 2013).

Urban surface energy balance models have been developed to represent the specific surface-atmosphere exchange in urban areas. The 3D building geometry directly influences the atmospheric flow (Moonen et al., 2012) in the urban roughness sub-layer whose depth is about 2-5 times the characteristic building height (Roth, 2000). It also leads to the interception of solar radiation and the trapping of infrared radiation. The latent heat flux in urban areas is usually lower than in rural areas owing to

less daytime evapotranspiration by vegetation, while the storage heat flux has a larger daily amplitude due to the high heat storage in construction materials. Human activities within urban areas serve as an additional source of heat and moisture (Sailor, 2011). These differences in the surface balances between urban and rural areas are responsible for the specific urban climate characterised by higher (nocturnal) air temperature (Arnfield, 2003), modified humidity (Unger, 1999), or altered precipitation (Shepherd, 2005).

Given the high relevance of the urban climate for the meteorological and climatological impact on humans and infrastructures, a variety of urban surface energy balance models has been developed (Masson 2006, Garuma 2018). Masson (2006) identified different categories: The empirical models are calibrated using observations; The modified vegetation models represent the specifics of urban areas by altering the physical properties of the flat surface; The single-layer and multi-layer Urban Canopy Models (UCMs) consider the 3D geometry of the buildings in a simplified way and solve the surface energy balance for the

roof, walls, and ground by taking into account their different physical characteristics, orientation and position. For the single-layer UCMs (Masson 2000, Kusaka et al. 2001), the first atmospheric model level is placed at the top of the urban roughness sublayer. The buildings receive the meteorological forcing from the first atmospheric model level only. The surface of the atmospheric model is located at roof level, the air volume below the characteristic building height (urban canopy layer) is therefore located below the surface of the atmospheric model. This way, only the lowest level of the atmospheric model is

directly influenced by the urban surface fluxes. For the multi-layer UCMs (Kondo and Liu 1998, Ca et al. 1999, Ca et al. 2002, Martilli et al. 2002), the buildings are immersed in the atmospheric model and receive the meteorological forcing from several atmospheric model levels. The effect of the buildings is taken into account in the atmospheric model by a drag force reducing the wind speed, a term representing the production of turbulent kinetic energy due to the buildings, a change in the turbulent mixing and dissipation length scale, and sometimes even considering explicitly the volume occupied by the buildings.

The single-layer UCMs are easier to couple with atmospheric models than the multi-layer UCMs since only minor modifications of the atmospheric model equations are required. The use of single-layer UCMs is justified for the historical European low- to mid-rise cities and at model resolutions down to 1 km (Trusilova et al., 2016). This is the resolution of the current operational limited area Numerical Weather Prediction (NWP) models. The new generation of NWP models shall be able to operate at down to 100 m horizontal resolution (Barlow et al., 2017) and take into account a larger variety of urban morpholo-

gies such as the high-rise Asian megacities. Increasing the vertical resolution can be useful to obtain more reliable near-surface diagnostics like air temperature and humidity at 2 m a.g.l., which could be calculated based on the prognostic model variables





instead of interpolating the simulated values between the first atmospheric level and the surface. Hamdi and Masson (2008) introduced a 1D column model in the UCM Town Energy Balance (TEB) to calculate the vertical profiles of the meteorological variables in the urban canopy layer, hereafter denoted with Surface Boundary Layer (SBL) scheme. This is a step towards better

near-surface diagnostics and obtaining more precise meteorological forcing for the walls, the impervious urban surfaces on the ground, and the urban vegetation. However, such an SBL scheme cannot take into account the process of advection in the urban canopy layer (e.g. from an urban park towards an adjacent densely built area). This deficiency has a larger effect for high-rise and heterogeneous cities than for homogeneous low- to mid-rise cities. A notable previous work to make up for this deficiency is the one of Chen et al. (2011), who coupled the multi-layer Building Effect parametrisation (BEP) to the Weather Research

and Forecasting model WRF (Skamarock et al., 2008), but this strategy is yet an exception in the world of LSM-UCMs.

## 1.2  Present study

The present study introduces a multi-layer coupling between the TEB, which is included in the LSM SURFEX and the research mesoscale atmospheric model Meso-NH (Lafore et al. 1998, Lac et al. 2018). SURFEX uses a tile-approach and distinguishes the four main surface types oceans (Voldoire et al., 2017), lakes (Salgado and Le Moigne, 2010), natural land surfaces (Noil-

han and Planton, 1989), and urban areas (Masson, 2000). SURFEX is the LSM used by various European NWP models like AROME (Seity et al., 2011), ALARO and ALADIN (Termonia et al., 2018), and the CNRM Earth System Model (Séférian et al., 2019). Given the previous areas of application of SURFEX-TEB in European cities, it is justifiable that it has been applied as a single-layer UCM only. The multi-layer coupling is developed here to prepare for the higher resolution NWP, and to enable the application of studies to quantify the benefit of climate change mitigation and adaptation measures for high-rise

cities.

The new (multi-layer) and old (single-layer) coupling is tested for the city of Hong Kong. The unique high-rise high-density urban environment, as well as the heterogeneous land cover and complex topography of this city have attracted much interest from the urban climate modelling community. Using the Fifth-Generation NCAR/PSU Mesoscale Model (MM5) and a simple bulk urban parametrisation, such as the Noah LSM (Chen and Dudhia, 2001), earlier studies focused on modelling the local

circulations and air quality during high air pollution episodes (Lam et al. 2006, Lo et al. 2007). Also using MM5-Noah LSM, Lin et al. (2007, 2009) investigated the effects of rapid urbanisation in the Pearl River Delta (PRD) region including Hong Kong on the regional climate at a model resolution of 3 km. A refinement in both the representation of urban surfaces and the model resolution has been made in later studies. Wang et al. (2014) conducted a systematic analysis of the seasonal variability in meteorological conditions influenced by land-use changes by employing WRF coupled to a single-layer UCM (Kusaka et al.,

2001). Recent studies adopt the more advanced multi-layer BEP (Martilli et al., 2002)-Building Energy Model (BEM; Sala-manca et al. 2009) scheme coupled to WRF to better consider the urban surface-atmosphere interactions. At a spatial resolution of 500 m, Wang et al. (2017, 2018) examined how tall buildings in Hong Kong could modify the boundary layer dynamics by introducing a new formulation of the drag coefficient as a function of building plan area density and implementing different air-conditioning systems. Making use of urban categories from the World Urban Database and Access Portal Tools initiative

(WUDAPT; Ching et al. 2018) and parameters derived from real building data, Wong et al. (2019) evaluated the uncertainties





due to different urban parameterisations and the precision of input data in urban climate simulations for Hong Kong. Instead of using an UCM, Dy et al. (2019) developed another approach to take into account the drag effects of urban surfaces at multiple atmospheric levels by modifying the Asymmetric Convective Model (ACM) planetary boundary layer scheme, which significantly improved the prediction of wind speed over the PRD region. The performance of the new multi-layer coupling between
Meso-NH and SURFEX-TEB introduced in the present study will be discussed against these studies in subsequent sections.

The main objectives of the present study are to introduce the new multi-layer coupling between Meso-NH and SURFEX-TEB, and to evaluate, for the single- and multi-layer coupling, the simulated near-surface meteorological variables air temperature, relative humidity, and wind, as well as building energy consumption under heat waves conditions. The present study is structured as follows. The new approach to couple Meso-NH and SURFEX-TEB is introduced in Section 2, the model configuration
and meteorological observations are presented in Section 3. Results are given in Section 4, discussion is made in Section 5, conclusions are drawn in Section 6.

## 2   New approach to couple Meso-NH and SURFEX-TEB

### 2.1   New multi-layer coupling approach

With the new multi-layer coupling approach (Figure 1), the buildings are immersed in the atmospheric model Meso-NH and
it is not required anymore to employ the SBL scheme to calculate vertical profiles for the meteorological parameters in the urban canopy layer. Instead, the meteorological forcing received by different urban facets is directly taken from the prognostic Meso-NH variables. Conversely, the momentum, heat and moisture fluxes from the building walls and roofs directly influence multiple atmospheric model levels. The influence of the buildings on the wind field is represented using a drag force approach and an additional production term in the prognostic equation for turbulent kinetic energy. The heat and moisture fluxes from the
walls and roofs to the atmosphere are injected at the corresponding model levels. The turbulent fluxes of momentum, sensible and latent heat from the urban impervious and pervious areas are directly influencing the lowest atmospheric model level. No change in the length scales for turbulent mixing and dissipation is made in Meso-NH. The physical equations of TEB remain unchanged. In particular, the geometric assumption of TEB that all buildings at grid point scale have the same height and are aligned along street canyons of infinite length employed for the calculation of the radiative exchanges is kept. Furthermore, the
walls are not discretised in the vertical direction, i.e. there is only one value for the prognostic wall temperature. The new multi-layer version of SURFEX-TEB is therefore simpler than the multi-layer WRF-BEP coupling presented by Chen et al. (2011), which allows one to take into account a variety of building heights at grid point scale and for which the vertical discretisation of the walls is imposed by the atmospheric model's grid. The multi-layer coupling introduced here keeps the simpler geometry of TEB. The advantage of the simpler multi-layer coupling is to represent the most important effect of the city - the fact that
the buildings are immersed in the atmosphere and not below the surface - while keeping the computational cost of the urban surface parametrisation low. The effect of the buildings on the prognostic Meso-NH variables is only considered between the surface and the mean building height to be consistent with the geometrical assumptions of TEB.

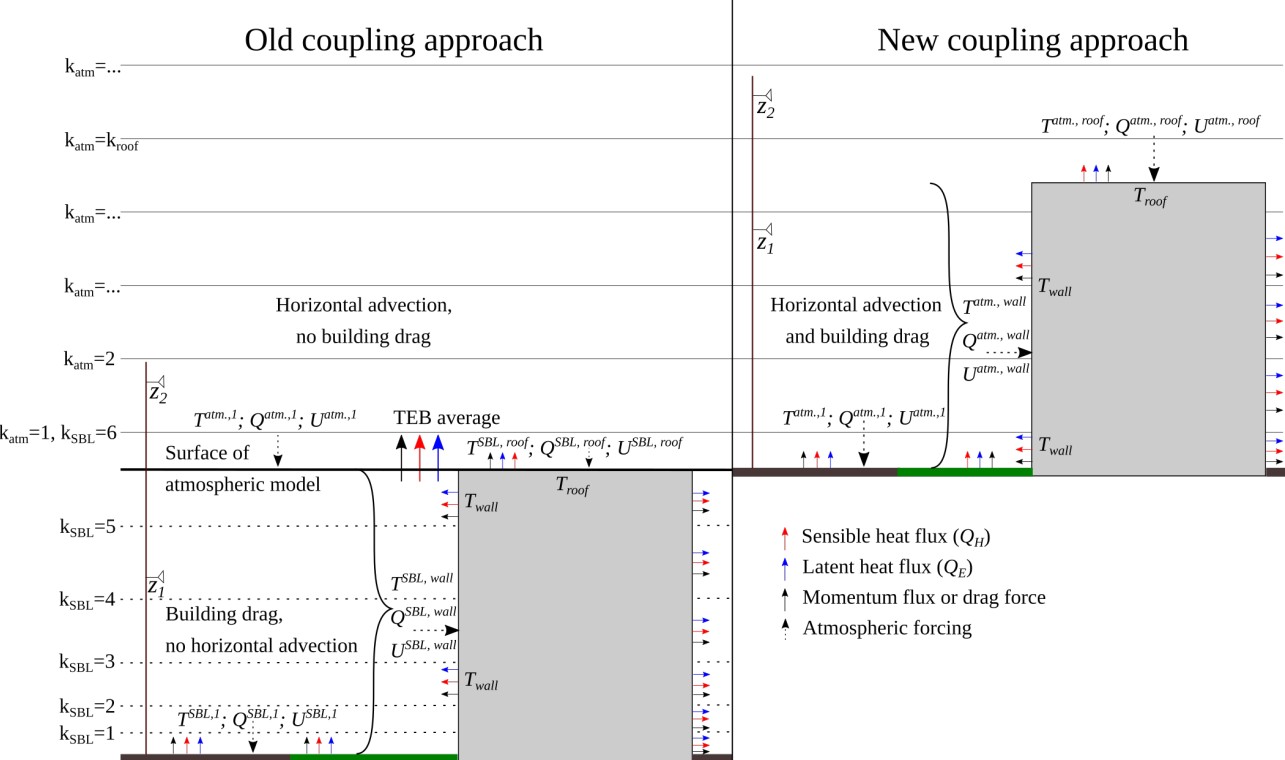

**Figure 1.** The old (single-layer) and new (multi-layer) approach for the coupling between Meso-NH and SURFEX-TEB. The two hypothetical wind anemometers at height above ground $z_1$ and $z_2$ represent at which height the model results for wind speed and direction are later compared with observations.

## 2.2 Equations

### 2.2.1 Modification of the Meso-NH equations

Meso-NH is a mesoscale anelastic nonhydrostatic atmospheric model whose basic equations are described in Lafore et al. (1998) and the most recent developments in Lac et al. (2018). The prognostic variables are the three velocity components $(u, v, w)$, the potential temperature $(\theta)$, the subgrid turbulent kinetic energy $(e)$, the mixing ratios of water vapour $(r_v)$ and other species like cloud droplets, and additional passive and reactive scalars. The model is written in flux form and the basic equations are discretised on a staggered Arakawa C grid, where meteorological and scalar variables are located in the center

of the grid cell and the momentum components on the faces of the cells. The coordinates follow the terrain. However, for simplification, the following equations are presented without reference to the terrain following coordinate system or the metric terms.

The friction exerted by the buildings on the horizontal wind components is taken into account using a drag force approach. The theoretical basis for this approach is explained in Raupach (1992). For highly three-dimensional flow over sparse roughness





elements (e.g. the buildings in the urban roughness sublayer), the total turbulent stress can be written as the sum of the stress on the roughness elements and the stress on the underlying surface. This approach assumes that the wake and drag properties of an isolated roughness element can be characterised by an effective shelter area and volume. This hypothesis is valid at low roughness density, but is unlikely to hold at high roughness density due to sheltering effects. For this reason, the drag force approach might yield uncertainties for high-density cities. The drag force approach translates into Equation 1 for the $u$

component (similar for the $v$ component).

$$\left.\frac{\partial(\rho_{d,ref}\,u)}{\partial t}\right|_{build} = -\rho_{d,ref}\left(c_d^{wall}d_{wall} + c_d^{roof}d_{roof}\right)u\,|\,\boldsymbol{u_{hor}}\,| \tag{1}$$

The dry air density of the reference state is denoted with $\rho_{d,ref}$. The horizontal wind speed ($|\,\boldsymbol{u_{hor}}\,|$) is calculated based on the prognostic $u$ and $v$ wind components (Equation 2).

$$|\,\boldsymbol{u_{hor}}\,| = \sqrt{u^2 + v^2} \tag{2}$$

The vertical frontal wall area density ($d_{wall}$) is calculated under the assumption that all buildings at grid point scale have the same height (Equation 3) to maintain coherence with the geometric assumption in TEB. A cylindrical building shape is assumed to calculate the frontal wall area density based on the wall to plan area ratio ($\lambda_w$). The real shape of the buildings is not taken into account, since this would require the definition of a large number of additional model input maps describing the frontal area density as a function of height and wind direction.

$$d_{wall}(z_k^m) = \begin{cases} \frac{\lambda_w}{\pi H_{bld}} & for \quad z_{k+1}^w < H_{bld} \\ \left(\frac{H_{bld}-z_k^w}{z_{k+1}^w - z_k^w}\right)\frac{\lambda_w}{\pi H_{bld}} & for \quad z_k^w < H_{bld} <= z_{k+1}^w \\ 0 & else \end{cases} \tag{3}$$

The grid point average building height is denoted with $H_{bld}$. The height above ground of the $k^{\text{th}}$ model level is $z_k^m$, the height above ground of the interfaces between the model levels is $z_k^w$.

The roofs are assumed to be horizontal. The vertical density of horizontal roofs ($d_{roof}$) is calculated following Equation 4.

$$d_{roof}(z_k^m) = \begin{cases} \frac{\lambda_p}{z_{k+1}^w - z_k^w} & for \quad z_k^w < H_{bld} <= z_{k+1}^w \\ 0 & else \end{cases} \tag{4}$$

The drag coefficient for the vertical walls ($c_d^{wall}$) is set to 0.4 since this corresponds to the value from wind tunnel studies reported by Raupach (1992) for cubes - a type of obstacles similar to actual buildings. This value has also been used by Martilli et al. (2002), Hamdi and Masson (2008), and Dy et al. (2019). The same formulation for the building drag, but different values for $c_d$ have been used by Uno et al. (1989) (0.1) or Oleson et al. (2008) (0.6). Santiago and Martilli (2010) used obstacle-resolving model simulations as a reference to determine uncertain parameters for UCMs. They found that a value of 0.4 for

$c_d$ led to too high wind speed in the urban canopy layer and instead propose a different formulation for the building drag that depends on the turbulent and spatial wind speed fluctuations. This new formulation performs better than $c_d = 0.4$, but would





require the introduction of additional diagnostic variables in the model. Its potential for improvement might be tested in future studies.

The drag coefficient due to the roofs is calculated similar to Hamdi and Masson (2008) following Equations 5 and 6.

$$c_d^{roof} = \left( \frac{u_*^{roof}}{\mid \boldsymbol{u_{hor}}(\boldsymbol{z_{k,roof}^m}) \mid} \right)^2 \tag{5}$$

$$u_*^{roof} = \frac{\kappa \mid \boldsymbol{u_{hor}}(\boldsymbol{z_{k,roof}^m}) \mid}{ln\left( \frac{(z_{k,roof}^m - H_{bld})}{z_{0,m}^{roof}} \right)} \tag{6}$$

$z_{k,roof}^m$ is the height above ground of the level, at least 0.5 m above the roof. The von Kármán constant ($\kappa$) is 0.4, the momentum roughness length of the roof ($z_{0,m}^{roof}$) is assumed to be 0.15 m. Atmospheric stability is not taken into account in the calculation of the friction due to the roofs; it is assumed that the strong wind shear close to the roofs dominates the effects due to buoyancy.

The production of subgrid turbulent kinetic energy ($e$) due to the wind shear close to the buildings walls and roofs is considered in a similar manner as in Martilli et al. (2002), Chin et al. (2005), and Hamdi and Masson (2008), following Equation 7.

$$\left. \frac{\partial(\rho_{d,ref}\, e)}{\partial t} \right|_{build} = \rho_{d,ref} \left( c_d^{wall} d_{wall} + c_d^{roof} d_{roof} \right) \mid \boldsymbol{u_{hor}} \mid^3 \tag{7}$$

The tendencies of potential temperature and water vapour mixing ratio due to the sensible ($Q_H^{wall}$, $Q_H^{roof}$) and latent ($Q_E^{wall}$, $Q_E^{roof}$) heat fluxes from the walls and the roofs towards the atmosphere are calculated following Equations 8 and 9.

$$\left. \frac{\partial(\rho_{d,ref}(z_k^m)\, \theta(z_k^m))}{\partial t} \right|_{build} = \begin{cases} \frac{Q_H^{wall}}{C_p H_{bld}} & for \;\; z_{k+1}^w < H_{bld} \\ \left( \frac{H_{bld} - z_k^m}{z_{k+1}^w - z_k^w} \right) \frac{Q_H^{wall}}{C_p H_{bld}} + \frac{Q_H^{roof}}{C_p(z_{k+1}^w - z_k^w)} & for \;\; z_k^w < H_{bld} <= z_{k+1}^w \\ 0 & else \end{cases} \tag{8}$$

$$\left. \frac{\partial(\rho_{d,ref}(z_k^m)\, r_v(z_k^m))}{\partial t} \right|_{build} = \begin{cases} \frac{Q_E^{wall}}{L_i H_{bld}} & for \;\; z_{k+1}^w < H_{bld} \\ \left( \frac{H_{bld} - z_k^m}{z_{k+1}^w - z_k^w} \right) \frac{Q_E^{wall}}{L_i H_{bld}} + \frac{Q_E^{roof}}{L_i(z_{k+1}^w - z_k^w)} & for \;\; z_k^w < H_{bld} <= z_{k+1}^w \\ 0 & else \end{cases} \tag{9}$$

Turbulent fluxes are in $\mathrm{Wm^{-2}}$ of total horizontal plan area of the grid point. They are calculated in the physical routines of TEB with respect to the potential temperature. The specific heat capacity of dry air ($C_p$) is 1005 $\mathrm{Jkg^{-1}K^{-1}}$, the specific heat $L_i$ is 2.5008E+6 $\mathrm{Jkg^{-1}}$ for evaporation and 2.8345E+6 $\mathrm{Jkg^{-1}}$ for sublimation.

### 2.2.2 Coupling between Meso-NH and SURFEX-TEB

The coupling between Meso-NH and SURFEX-TEB is technically modified such that SURFEX-TEB can receive the forcing from the first to the $NC^{\mathrm{th}}$ ($1 : NC$) Meso-NH level. For the sake of simplicity, the horizontal dimensions of the Meso-NH





variables are not explicited in the equations. The horizontal wind speed ($\mid \boldsymbol{u_{hor}} \mid$) is calculated based on the prognostic $u$ and $v$ wind components (Equation 10).

$$\mid \boldsymbol{u_{hor}(z^m_{1:NC})} \mid = \sqrt{u(z^m_{1:NC})^2 + v(z^m_{1:NC})^2} \tag{10}$$

The air temperature ($T$) is calculated based on the prognostic potential temperature ($\theta$) and the Exner function ($\Phi$) following

$$T(z^m_{1:NC}) = \theta(z^m_{1:NC})\Phi(z^m_{1:NC}) \tag{11}$$


$$\Phi(z^m_{1:NC}) = \left(\frac{p(z^m_{1:NC})}{p_0}\right)^{\frac{R_d}{C_p}} \tag{12}$$

where $p$ is the diagnostic absolute pressure. The specific gas constant for dry air ($R_d$) is 287.01 $\mathrm{Jkg^{-1}K^{-1}}$, the reference pressure ($p_0$) is 1.0E+5 Pa. The absolute humidity ($q$) is calculated based on the prognostic mixing ratio of water vapour ($r_v$) following

$$q(z^m_{1:NC}) = r_v(z^m_{1:NC}) \, \rho_{d,ref}(z^m_{1:NC}) \tag{13}$$

and the density of the moist air ($\rho$) is given by

$$\rho(z^m_{1:NC}) = \frac{p(z^m_{1:NC})}{R_d T(z^m_{1:NC})} \frac{1 + (R_d/R_v)r_v(z^m_{1:NC})}{1 + r_v(z^m_{1:NC})} \tag{14}$$

where the specific gas constant for water vapour ($R_v$) is 461.5 $\mathrm{Jkg^{-1}K^{-1}}$.

The height of the atmospheric forcing level used by SURFEX-TEB ($z_{k,forc.}$) needs to be calculated based on the height of
the Meso-NH levels ($z^m_k$). It is ambiguous whether this forcing height should be the distance of the atmospheric level to the potentially inclined surface (inclination angle $\alpha$) or the vertical height above the surface. It is assumed that for katabatic winds located in the first few meters above ground level (a.g.l.), the distance to the surface is the most relevant, whereas for the other processes it will be the vertical height above the surface. Therefore, the forcing height is defined as the shortest distance between the model level and the surface in the lowest 5 m vertical distance to the surface, and as the vertical distance at or
above 20 m vertical distance to the surface (Equation 15). A linear transition is applied in between (Equations 15 and 16).

$$z_{k,forc.} = f_k z^m_k + (1 - f_k)z^m_k cos(\alpha) \tag{15}$$

$$f_k = min\left(1.0, \frac{max(z^m_k - 5.0, 0.0)}{15.0}\right) \tag{16}$$

### 2.2.3   Modification of the SURFEX-TEB equations

The multi-layer coupling of TEB is technically enabled by a logical switch which deactivates the prognostic equations of the SBL scheme of Hamdi and Masson (2008) and instead at each time step fills the SBL scheme's prognostic variables with the





corresponding Meso-NH variables. With this implementation it is easy to switch between the single-layer and the multi-layer coupling. The meteorological forcing for the impervious surfaces such as roads ($imp.$) and the low urban vegetation ($lveg.$) is taken from the first Meso-NH level following

$$U_{forc.}^{imp./lveg.} =\mid \boldsymbol{u_{hor}}(1)\mid ; \; T_{forc.}^{imp./lveg.} = T(1) ; \; Q_{forc.}^{imp./lveg.} = \frac{q(1)}{\rho(1)} \tag{17}$$

where $Q$ denotes the specific humidity, $U$ the wind speed, and the height of the forcing is given by

$$z_{forc.}^{imp./lveg.} = z_{1,forc.} \tag{18}$$

The meteorological forcing for the roof (Equation 19) is taken from the closest Meso-NH level, but at least 0.5 m above the roof ($k_{roof}$).

$$U_{forc.}^{roof} =\mid \boldsymbol{u_{hor}}(k_{roof})\mid ; \; T_{forc.}^{roof} = T(k_{roof}) ; \; Q_{forc.}^{roof} = \frac{q(k_{roof})}{\rho(k_{roof})} \tag{19}$$

The height of the forcing above the roof is

$$z_{forc.}^{roof} = z_{k_{roof},forc.} - H_{bld} \tag{20}$$

Since in TEB the walls are not vertically discretised, there is only one value for the prognostic wall temperature at grid point scale, hence the average value of the meteorological forcing variables is calculated for all Meso-NH levels intersecting the walls (Equations 21 to 23).

$$U_{for}^{wall} = \frac{1}{H_{bld}} \sum_{k=1}^{k_{roof}} \begin{cases} \mid \boldsymbol{u_{hor}}(k)\mid (z_{k+1}^{w} - z_{k}^{w}) & for \quad z_{k+1}^{w} <= H_{bld} \\ \mid \boldsymbol{u_{hor}}(k)\mid (H_{bld} - z_{k}^{w}) & for \quad z_{k}^{w} < H_{bld} < z_{k+1}^{w} \end{cases} \tag{21}$$

$$T_{for}^{wall} = \frac{1}{H_{bld}} \sum_{k=1}^{k_{roof}} \begin{cases} T(k)(z_{k+1}^{w} - z_{k}^{w}) & for \quad z_{k+1}^{w} <= H_{bld} \\ T(k)(H_{bld} - z_{k}^{w}) & for \quad z_{k}^{w} < H_{bld} < z_{k+1}^{w} \end{cases} \tag{22}$$

$$Q_{for}^{wall} = \frac{1}{H_{bld}} \sum_{k=1}^{k_{roof}} \begin{cases} \frac{q(k)}{\rho(k)}(z_{k+1}^{w} - z_{k}^{w}) & for \quad z_{k+1}^{w} <= H_{bld} \\ \frac{q(k)}{\rho(k)}(H_{bld} - z_{k}^{w}) & for \quad z_{k}^{w} < H_{bld} < z_{k+1}^{w} \end{cases} \tag{23}$$

The sensible and latent heat fluxes from the roof, walls, impervious and pervious surfaces to the air in the street canyon are then calculated with the same formulas that are detailed in Hamdi and Masson (2008) and Lemonsu et al. (2012).

## 2.3 Uncertainties of the multi-layer coupling between Meso-NH and SURFEX-TEB

Various uncertainties remain in the presented multi-layer coupling, which could be addressed in future studies:





– The variation in building height at grid point scale is neglected. This might lead to too high wind speed values above the average building height and too low values below.

  – Building drag only depends on the local value of the frontal wall area density, which is assumed to be isotropic. The building shape and orientation, which could potentially lead to a directional variation of the drag coefficient is not taken into account. Furthermore, channelling in the streets can lead to changes of the drag coefficient (Santiago et al. 2013,

and Simón-Moral et al. 2014).

  – In contrast to numerous previous studies, the turbulent mixing and dissipation length scales are not modified in the urban environment. The mixing length scales for urban areas proposed by Santiago and Martilli (2010) have been tested (not shown) and lead to a deterioration of the model results. However, it cannot be excluded that alternative formulations for turbulent length scales in the urban environment might improve results.

– The potential influence of the thermal stratification on the building drag is neglected. Based on obstacle-resolving modelling, Santiago et al. (2014) and Simón-Moral et al. (2017) found that the building drag increases for unstable stratification due to the enhanced vertical mixing.

  – The volume occupied by the buildings is neglected in the Meso-NH equations, i.e. the building heat and moisture fluxes are injected into the entire volume of the grid cell. This might lead to greater uncertainties, the denser the cities are.

– The turbulent surface fluxes on the horizontal urban facets like roofs and impervious surfaces are calculated using the Monin-Obukhov Similarity Theory (MOST), which is questionable since the surface characteristics and the flow are not horizontally homogeneous (Martilli et al., 2002).

  – The drag force due to high urban vegetation is not considered. It could be introduced similar to Santiago et al. (2019), Redon et al. (2020), or Krayenhoff et al. (2020).

– Radiation is only coupled at the surface. It is therefore neglected that high-rise buildings might receive a considerably different amount of radiation than the surface (e.g. due to urban air pollution or fog) and that they emit longwave radiation not only into the first atmospheric model level. A more coherent treatment of radiative exchanges between the urban canopy layer and the free atmosphere will soon become possible thanks to recent developments (Hogan 2019a and Hogan 2019b), but could not be included in the present study.

– The rain and snow rate is taken from the surface level of the atmospheric model. It is therefore neglected that precipitation intercepted by high-rise buildings might be different from surface level precipitation. Furthermore, the precipitation is only intercepted by the roofs and the ground, and not by the walls.

  – The mixing ratios of chemical substances and carbon dioxide are coupled only at the first atmospheric level. Multi-layer coupling may be introduced, e.g. to take into account for emissions due to high chimneys. This might improve model

results especially during situations with stable atmospheric stratification.





## 3   Model setup for evaluation of different coupling approaches

### 3.1   Selected meteorological situations

With relevance for heat-health impact assessment (Wang et al., 2019) and heat stress mitigation (Aflaki et al., 2017), two prolonged high temperature events – 1 to 8 September 2009 and 17 to 31 May 2018 – are selected for model evaluation in

the present study. During these selected periods, the HKO recorded 8 and 15 consecutive Very Hot Days (daily maximum air temperature of $33°C$ or above measured at the HKO Headquarters station), respectively, with the latter breaking the record set in May 1963 by a large margin (HKO, 2018). Under the influence of a high pressure system over the northern part of the South China Sea, both periods experienced fine, sunny conditions with long duration of sunshine and a lack of precipitation. These characteristics correspond to those of the typical heat waves occurring in southern China, which are found to be attributable

to the westward displacement of the western North Pacific subtropical high pressure system, causing an anomalously dry and warm anticyclonic flow (Luo and Lau, 2017). However, the synoptic wind flow over Hong Kong differs for the two selected periods, with prevailing winds from the east and southwest for the heat waves in September 2009 (HW2009) and May 2018 (HW2018), respectively. The dominant south-westerly wind during HW2018 coincides with the typical summer prevailing wind direction in Hong Kong (Ng et al., 2012), making it a representative reference simulation period for the subsequent

investigation of future development and mitigation scenarios.

In order to evaluate the modelled anthropogenic heat flux due to the buildings against inventory data available only on monthly time scale, a simulation using the new multi-layer coupling is also conducted for the entire month of May 2018, which was characterised by a mean monthly air temperature 2.4 K above the long-term (1981-2010) normal of $28.3°C$.

### 3.2   Model configuration

For modelling the selected meteorological situations, Meso-NH is employed to downscale the high-resolution operational forecast analyses from the European Centre for Medium-Range Weather Forecasts (ECMWF) Integrated Forecasting System via three intermediate domains (D1, D2, D3) to a domain covering major parts of Hong Kong at 250 m horizontal resolution (D4), and a 125 m resolution domain covering Hong Kong Island and Kowloon Peninsula (D5). Table 1 summarises the employed physical parameterisations; the delimitation of the model domains is displayed in Figure 2a. Only the hourly model

outputs for D4 and D5 are analysed.

Meso-NH is coupled with SURFEX (Masson et al., 2013) to solve the surface energy budget; more details on the tested coupling approaches are given in Section 3.4. The urban vegetation located in the street canyon is represented with the approach of Lemonsu et al. (2012). The energy budget of a representative building at district scale is calculated by a building energy model (Bueno et al. 2012, Pigeon et al. 2014). Information on the urban form and function of Hong Kong is taken from

Kwok et al. (2020, accepted by Theoretical and Applied Climatology). This dataset includes maps at 100 m resolution of the urban morphology parameters (e.g. $H_{bld}$, $\lambda_p$, $\lambda_w$), and a map at 100 m resolution of the dominant building type taken from an ensemble of 18 typical buildings (archetypes) in Hong Kong defined by Kwok et al. (2020). For each of the archetypes, Kwok et al. (2020) provide a description of the construction materials and their physical properties, the temporal evolution




**Table 1.** Physical parameterisations employed for the Meso-NH simulations.

| Domain | Horizontal resolution [km] | Time step [s] | Parametrisation of deep convection | Parameterisation of shallow convection and dry thermals | Mixing length calculation |
|--------|----------------------------|---------------|------------------------------------|----------------------------------------------------------|---------------------------|
| D1 | 8 | 20 | Kain and Fritsch (1990) | Pergaud et al. (2009) | Bougeault and Lacarrère (1989) |
| D2 | 2 | 10 | None | Pergaud et al. (2009) | Bougeault and Lacarrère (1989) |
| D3 | 1 | 5 | None | Pergaud et al. (2009) | Bougeault and Lacarrère (1989) |
| D4 | 0.25 | 1.7 | None | None | Deardorff (1980) |
| D5 | 0.125 | 0.8 | None | None | Deardorff (1980) |

of the internal heat release due to electrical appliances and domestic warm water, and of the setpoint temperature for air

conditioning. The anthropogenic heat flux due to air conditioning is simulated by the building energy model as a function of these input data. The total simulated building-related anthropogenic heat flux is the sum of the contributions from the electrical appliances, lighting, cooking, domestic warm water, and air conditioning of buildings. The anthropogenic heat flux due to traffic is neglected since it is about a factor of four lower than the anthropogenic heat flux due to the buildings. The dataset describing Hong Kong represents the city in 2018 and is therefore optimal for the simulation of HW2018 and might slightly

overestimate urbanisation in some areas for HW2009. The land cover parameters for the rural areas are taken from the 1 km resolution Ecosystem Climate Map (ECOCLIMAP-I) database (Masson et al. 2003, Champeaux et al. 2005). Daily values of the Sea Surface Temperature (SST) have been taken from the Global Ocean 1/12° Physics Analysis and Forecast provided by the European Union Copernicus Marine Service Information. The daily SST values are interpolated linearly in time. Aerosol optical depth is set to the spatially and temporally uniform value of 0.1.

## 3.3 Meteorological observations and building anthropogenic heat flux inventory for model evaluation

### 3.3.1 Meteorological observations

Near-surface meteorological observations obtained from the Hong Kong Observatory (HKO) are used for model evaluation. Hong Kong has a well-established network of more than 50 Automatic Weather Stations (AWS), of which 34 are located within the two innermost model domains employed in the present study (Figure 2b, HKO 2020). Hourly observations of air tempera-

ture, relative humidity, wind speed and direction, and rainfall are available at 30, 19, 20, and 19 of these stations, respectively. Solar radiation is observed at stations KP and KSC. Thermometers and hygrometers are placed in Stevenson screens around 1 m a.g.l., and wind anemometer heights vary from 9 m to 42 m a.g.l.. Due to the complex terrain and heterogeneous land cover and urban settings in Hong Kong, model evaluation is particularly challenging as AWS are situated in a diverse range of environments, including urban parks surrounded by tall buildings (e.g. KP, HKP, KTG), vegetated rural areas (e.g. TYW,

KFB), piers (e.g. CP, SE1), mountain peaks (e.g. TMS, TC), outlying islands (e.g. WGL, CCH), and the rooftop of a high-rise building (CPH). Characteristics of station environments are therefore quantified in terms of artificial surface cover fractions

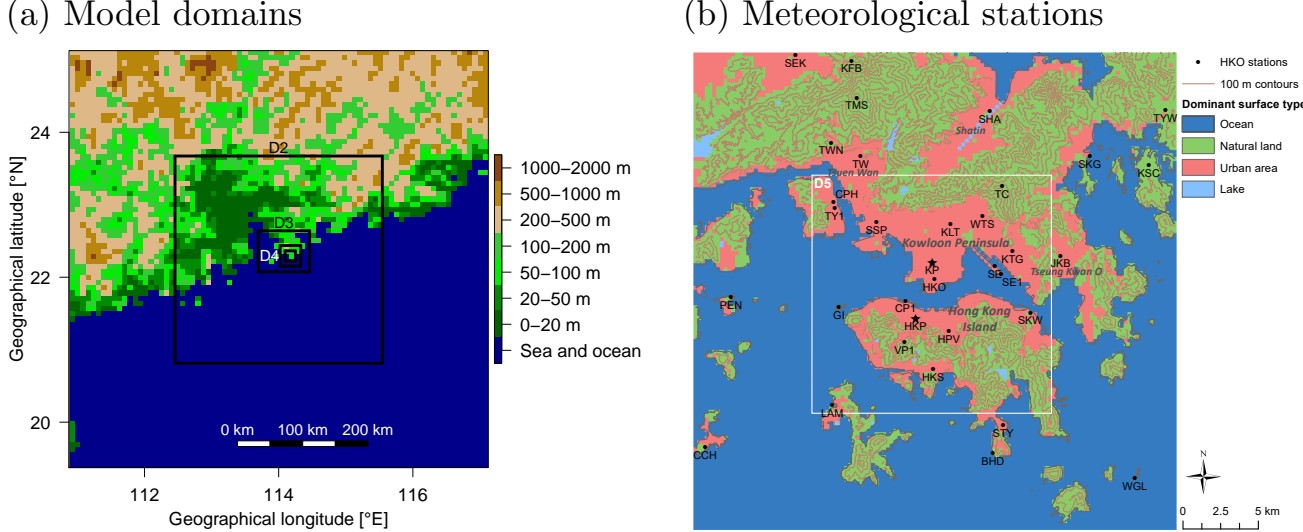

**Figure 2.** (a) The five nested Meso-NH model domains employed for the high-resolution simulation of the urban climate of Hong Kong. (b) Meteorological stations operated by the Hong Kong Observatory that are located within the model domains D4 and D5. Model results will be presented in more detail at the starred stations KP and HKP.

and average building height to facilitate a systematic evaluation of model output (Table A1). However, one should also bear in mind the uncertainties introduced by the averaging of surface and morphological parameters within model grids. Moreover, the authors observed through site visits that some measurements might be heavily influenced by obstacles close to the stations, such as buildings to the windward side of the station and tree canopies above the station (Figure 3), and thus affecting the representativeness of the observations.

In addition to the fixed network of meteorological stations, radiosoundings at the station King's Park (KP) are available for 0 and 12 UTC in September 2009 and 0, 6, 12, 18 UTC in May 2018. Meteorological data have been recorded every 2 s, which, for the ascent rate of 250 to 450 m/min corresponds to a vertical resolution of 8 to 15 m. The radiosoundings are used to evaluate the vertical profiles of the simulated meteorological parameters in the lower part of the urban boundary layer. Furthermore, local SST measurements are available twice a day (at 7 and 14 local time) within the harbour near the North Point Fire Station (around 3 km northwest of station SKW in Figure 2b) and hourly in the open coastal waters adjacent to station WGL.

### 3.3.2   Building anthropogenic heat flux inventory

The monthly total electricity and gas consumption for the entire Hong Kong is published for different sectors by the Hong Kong Census and Statistics Department (HKC, 2018). The energy consumption of the domestic, commercial, and industrial sectors is used, and the energy consumption of the transport sector and for street lighting is excluded. It is assumed that buildings are the only contributors to energy consumption within the selected sectors, that all the consumed energy is released into the





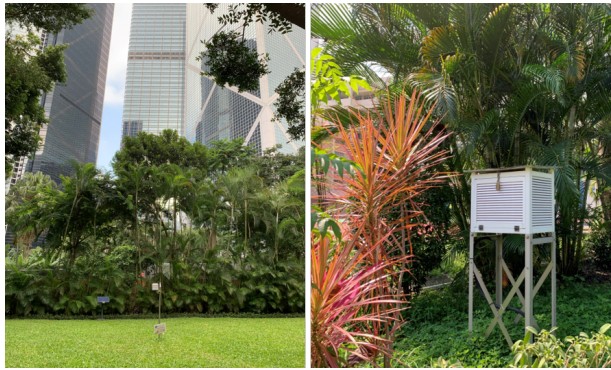

**Figure 3.** Environment of the meteorological stations HKP (left) and TW (right).

air, and that all buildings with the same use exhibit the same volumetric energy consumption. Under these assumptions, the inventory-based anthropogenic heat flux ($Q_{a,i}^{sec}$) due to building $i$ in sector $sec$ in Wm$^{-2}$ is calculated following Equation 24.

$$Q_{a,i}^{sec} = \frac{E_{monthly}^{sec}}{N \; A_{tot}^{sec}} \; \frac{V_i}{V_{mean}^{sec}} \tag{24}$$

$E_{monthly}^{sec}$ is the total monthly sector wide energy consumption in J, $A_{tot}^{sec}$ the total building surface area of sector $sec$ in m$^2$, $V_i$ the volume of building $i$, $V_{mean}^{sec}$ the sector mean building volume, and $N = 2678400$ the number of seconds in May. Heat fluxes are then aggregated to the model grid resolution and Figure 12 shows the resulting map of anthropogenic heat flux due to building energy consumption in May 2018. Although this inventory is subject to uncertainties, the spatial pattern is reasonably realistic and similar to that estimated by Wong et al. (2015) using remote sensing methods, except for an underestimation in certain areas like the airport and container terminals, where energy-intensive activities do not take place within buildings.

### 3.4 The tested coupling approaches

The approaches to couple Meso-NH and SURFEX-TEB tested in the present study are described. For the evaluation, the model level with the height a.g.l. closest to that of the meteorological station needs to be selected. In urban areas, this can be very different between the single- and multi-layer coupling approach, since with the single-layer coupling approach, the buildings are located below the surface of the atmospheric model, whereas with the multi-layer coupling approach the buildings are immersed in the atmosphere (Figure 1). In the horizontal dimension, for all coupling approaches, the model grid point with the shortest distance to the meteorological station is taken. The investigated coupling approaches are:

– CLASSICAL corresponds to the classical single-layer approach to couple Meso-NH and SURFEX-TEB used so far. The vertical grid of Meso-NH is relatively coarse; the first level is placed at 10 m a.g.l., the vertical atmospheric grid size near the surface is 20 m and increases by 10% every model level to a maximum of 500 m. The aerodynamic roughness length of the urban area is $z_{0,m} = 0.1 \, H_{bld}$. The SBL scheme of Hamdi and Masson (2008) is used to simulate vertical profiles of the meteorological variables in the urban canopy layer. The SBL scheme has six levels. The first two levels are located



at 0.5 m and 2 m a.g.l., the other four levels are placed such that the height a.g.l. of the sixth level matches with the height a.g.l. of the first atmospheric level (Figure 1). For the rural areas, a similar SBL scheme with six levels introduced by Masson and Seity (2009) is employed. For the evaluation of air temperature and humidity in urban and rural areas, which is observed in around 1 m a.g.l., the simulated values from the first two levels of the urban or rural SBL scheme are linearly interpolated. The wind measurements might be, depending on the height of the anemometer for each station, located below or above the highest SBL level. If the height of the wind anemometer is below the highest SBL level, the simulated values from the two closest SBL levels are interpolated linearly. Otherwise, the Meso-NH levels are taken.

– NEW corresponds to the new multi-layer coupling. The surface of the Meso-NH model corresponds to the physical surface, including in the urban area. No SBL scheme is required to calculate vertical profiles of the meteorological parameters in the urban canopy layer. Since the drag force due to the building walls and roofs is considered directly in the atmospheric model, the aerodynamic roughness length is set to 0.05 m in urban areas to represent the roughness of the urban impervious and pervious ground surfaces. Due to the modified coupling approach, it is possible to refine the vertical grid of Meso-NH. The first scalar model level is placed at 1 m a.g.l., the vertical resolution near the surface is 2 m and increases by 15% with increasing distance to the surface to a maximum of 500 m. For the high rural vegetation (e.g. forests), the approach of Aumond et al. (2013) is used to represent the drag force it exerts on the wind. As a consequence, the rural SBL scheme is also deactivated. For all meteorological variables, the prognostic Meso-NH variables from the two model levels nearest to the station height are linearly interpolated to compare to observations.

– SURFFLUX is similar to NEW, except that the fluxes of heat and moisture from the building walls and roofs to the atmosphere are released at the surface of Meso-NH. This coupling approach is of interest since the coupling of temperature and humidity in the new coupling approach is explicit, which would not be viable when using larger time steps (e.g. in an earth system model). This experiment can give hints of whether it is worthwhile to develop an implicit coupling for heat and moisture in the future.

## 4 Results

### 4.1 Time series of near-surface meteorological variables

#### 4.1.1 Explorative analysis at two urban stations

The simulated time series of all relevant meteorological variables are first presented in detail for D4 at the stations King's Park (KP), the urban station with the most comprehensive observational data and Hong Kong Park (HKP) with the highest buildings in the surrounding. KP is located in an urban park (about 500 m x 500 m) on a small hill at the heart of Kowloon Peninsula surrounded by buildings with a typical height of 30 m. HKP is located in a 8 ha park amid the high-rise high-density business district on the northern coast of Hong Kong Island. The high-rise buildings surrounding HKP have a large variety in building height, with an average of 100 m (Figure 3), and a large variability in building height. KP measures all relevant meteorological



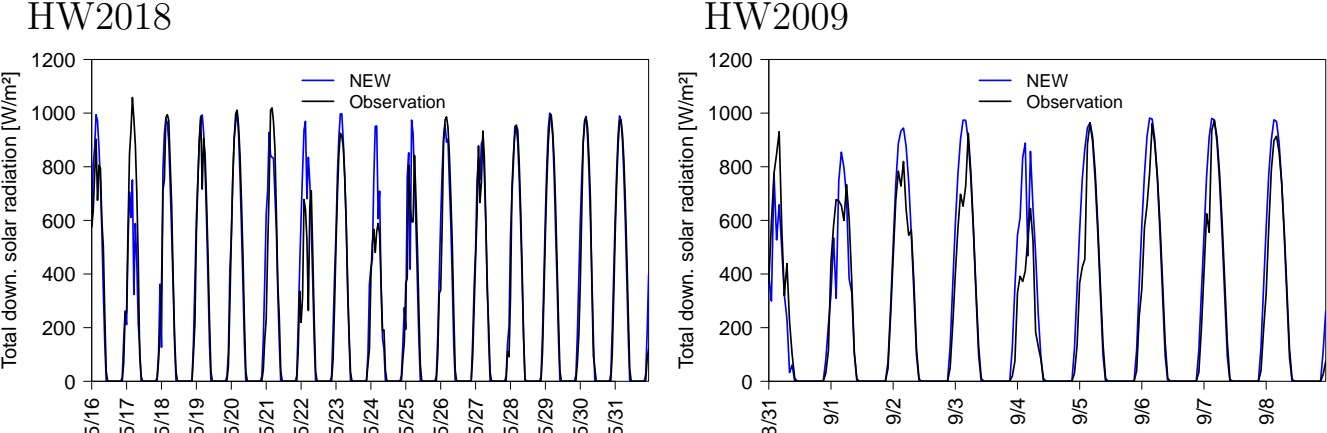

**Figure 4.** Time series (UTC) of simulated (NEW coupling approach, D4) and observed solar radiation at the station King's Park (KP).

parameters, HKP only near-surface air temperature.

Simulated and observed total downwelling solar radiation at station KP is displayed in Figure 4 for HW2018 and HW2009. Only the results for NEW are shown, since the different coupling approaches do not alter the simulated downwelling solar radiation in a relevant manner. Simulated solar radiation is very close to the observed on cloud-free days. This indicates that the selected value for the aerosol optical depth of 0.1 is appropriate for the selected heat waves. Due to the synoptic scale flow

from the south-west (HW2018) and east (HW2009), no air pollution is advected from China, which would have to be taken into account via a time dependent aerosol optical depth. Larger biases in downwelling solar radiation appear for days with observed clouds for which the model tends to overestimate the downwelling shortwave radiation (e.g. Sep. 1 to 4 2009 and May 22 to 24 2018). During HW2018 there are also two days during which too many clouds are simulated compared to the observations (May 17 and May 21). In summary, solar radiation is overestimated for both heat waves with a small bias of 10 $\mathrm{Wm}^{-2}$ for

HW2018, and a larger bias of 42 $\mathrm{Wm}^{-2}$ for HW2009.

The time series of air temperature and relative humidity at 1 m a.g.l. and wind speed and direction at 25 m a.g.l. are displayed in Figure 5 (Figure 6) for HW2018 (HW2009) at KP. For HW2018, CLASSICAL leads to an overestimation of air temperature of 2 to 4 K in the daytime, which is nearly entirely corrected for NEW. The nighttime air temperature is simulated well with all coupling approaches. During the end of HW2018, both daytime and nighttime air temperature are overestimated for NEW,

whereas CLASSICAL also overestimates the amplitude of daily temperature variation. The results for SURFFLUX do not differ much from those for NEW, since there are only a few mid-rise buildings in the grid cell where KP is located. The release of the heat fluxes from the walls and roofs of these buildings at the surface therefore does not deteriorate the model results. Relative humidity in the nighttime is underestimated for all coupling approaches and, similar to air temperature, does not differ much between the different approaches. Daytime relative humidity is underestimated for CLASSICAL and better simulated

for NEW, but the differences between the coupling approaches are not as large as for air temperature. The simulated values



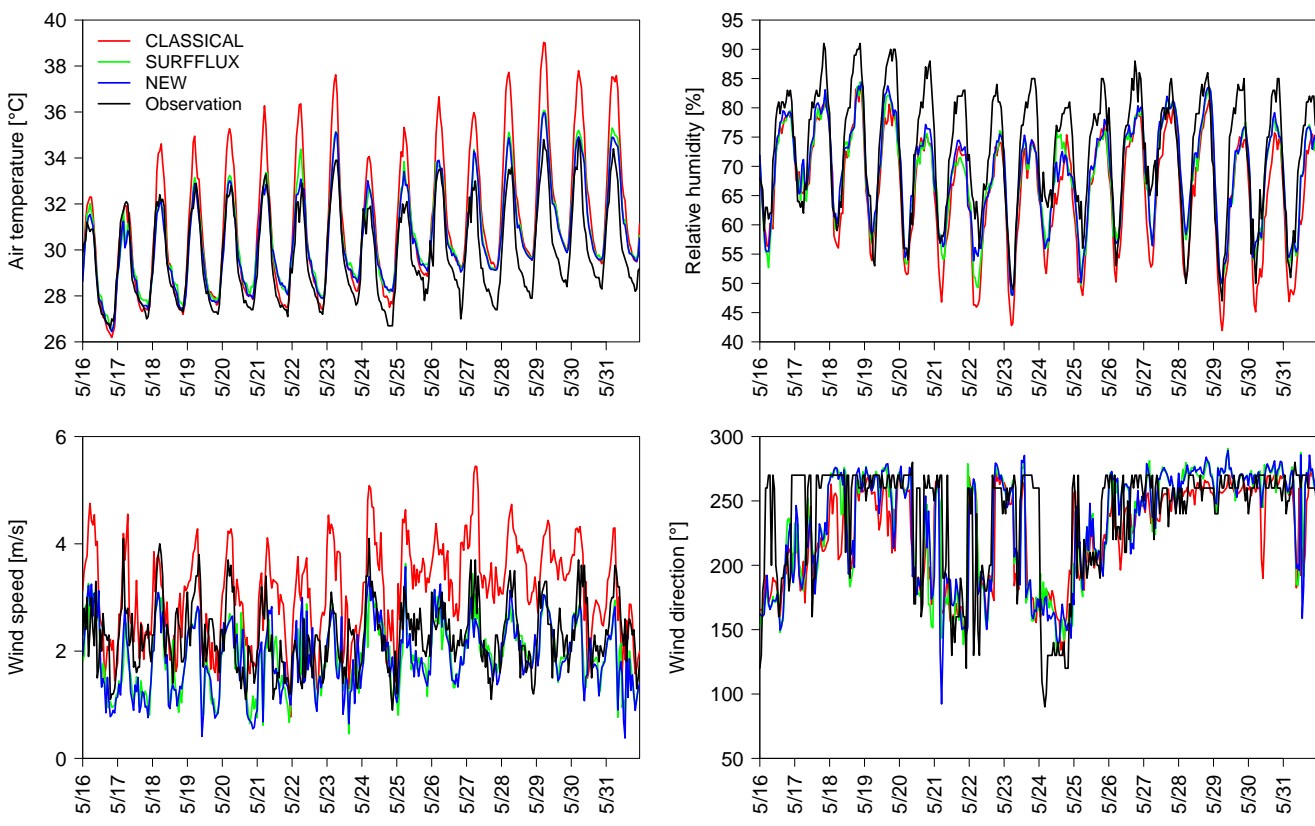

**Figure 5.** Time series (UTC) of simulated (D4) and observed meteorological variables at the station King's Park (KP) during HW2018.

of wind speed are too high for HW2018 and CLASSICAL, since the drag force due to the buildings is not considered in the atmospheric model. For NEW, the simulated wind speed values are reduced and agree better with observations, although they are slightly underestimated at the beginning of the heat wave. No relevant differences for wind speed are found between NEW and SURFFLUX.

Results for HW2009 differ from those for HW2018. Both the CLASSICAL and the NEW coupling approach lead to an overestimation of daytime air temperature, whereas the nighttime air temperature is well simulated. The fact that air temperature is overestimated can be explained by the too high values of simulated downwelling solar radiation. NEW performs better than CLASSICAL, but the differences are lower than for HW2018. This might be due to the different wind direction. For HW2018, air is advected from a very densely built environment west of the station, whereas for HW2009 it is advected from

a less densely built area east of the station. This could explain the lower difference between the two coupling approaches for HW2009. Relative humidity is equally underestimated for all coupling approaches, which is consistent with the overestimation of air temperature. Wind speed is overestimated for CLASSICAL and underestimated for NEW.

The main observed prevailing wind direction at station KP is west (east) for HW2018 (HW2009). This is very well represented

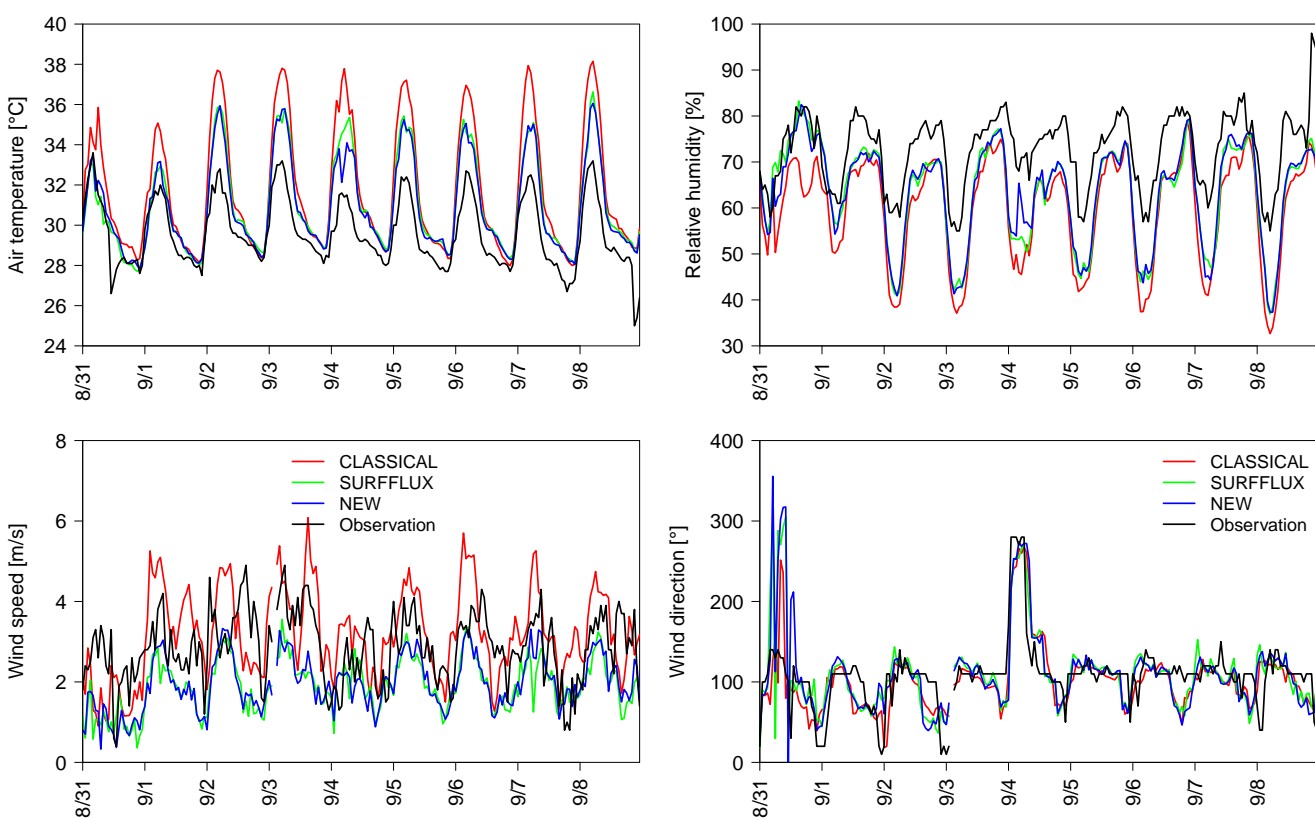

**Figure 6.** Same as Figure 5, but for HW2009.

in the model and there are only small differences between the coupling approaches. For HW2018, the wind direction observed

at station KP is different from the synoptic-scale wind direction (southwest to south) due to the circulation around Hong Kong Island in combination with sea breezes (Figure 10cd) Slight changes in the synoptic-scale wind direction from south-west to south-south-east can lead to a strong change in wind direction over Kowloon Peninsula from west to east since the circulation around Hong Kong Island changes direction. This takes place twice during HW2018 (May 21 and 22) and May 24, which is represented by the model, with the exception that the onset of the circulation shift is 12 hours too early for May 24. The

easterly wind direction during HW2009 is reproduced by the model, the shift towards a westerly circulation on September 4 is captured. Interestingly, both observations and model display a shift towards north-easterly wind in the late evening, although not perfectly coherent in time and magnitude. This is due to the higher elevation in the north-east of the KP station than at the station itself, which influences the local circulation in the nighttime.

The time series of air temperature at 1 m a.g.l. for the station HKP, which is surrounded by high-rise buildings, are displayed

in Figure 7. The findings at this station are similar to KP, but exacerbated. The CLASSICAL coupling approach leads to an overestimation of the daytime air temperature by at least 4 K for both heat waves. The nighttime air temperature is acceptable



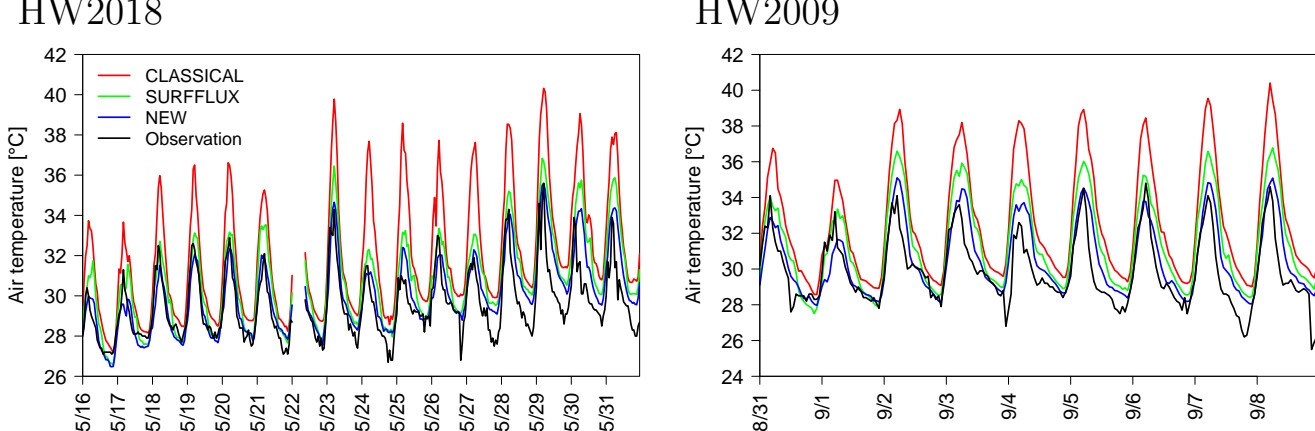

**Figure 7.** Time series (UTC) of simulated (D4) and observed near-surface air temperature at the station Hong Kong Park (HKP).

at the beginning of both heat waves and overestimated by 1 to 3 K at the end. This could be due to too high heat storage in the construction materials in the course of the heat wave due to the too high values of the simulated daytime air temperature in the urban canopy layer. Simulated and observed time series agree well for NEW during both heat waves. For SURFFLUX,

the simulated values are close to the observations in the nighttime, but in the daytime, the model performance is clearly worse than for NEW. The release of the large sensible heat fluxes from the building walls and roofs at the surface leads to a clear deterioration of model results in this high-rise high-density setting.

### 4.1.2 Model evaluation measures at all stations

The bias and root mean square error (rmse) of the simulated hourly time series of air temperature, relative humidity, and wind

speed at all available stations are displayed in Figure 8 (Figure 9) for HW2018 (HW2009) and D4; the Figures for D5 are given in the Appendix (Figures B1 and B2). Given the low value of the daily temperature amplitude of 5 K to 8 K during the simulated heat waves it is easy to obtain seemingly good values for the rmse compared to locations with continental or mid-latitude climates. It is therefore considered that values of the rmse >1.5 K denote a model result of unacceptable quality. With a similar reasoning, a rmse >10% for relative humidity is considered as an unacceptable model result. In the following

discussion, those stations with a building surface fraction larger than 0.1 or an average building height taller than 15 m within a circle of radius 250 m around the station are considered as urban. The other stations are considered as rural.

For CLASSICAL, the evaluation measures for air temperature are not acceptable for most urban stations. For HW2018, results are particularly bad at the stations HKO, HKP, HKS, JKB, STY, TWN, and TY1 with values of the rmse around or larger than 2.5 K. These are the stations located in urban parks surrounded by high-rise buildings (HKO, HKP) and the stations in

very heterogeneous areas with mid or high-rise buildings close to vegetated areas or the coast (HKS, JKB, STY, TWN, TY1). For NEW, bias and rmse are improved for all the urban stations, the rmse ranges mostly between 1 and 1.5 K, which is an





acceptable model result. The bias is positive for all urban stations, which might be due to the slight overestimation of the downwelling solar radiation or too high SST. Evaluation measures for SURFFLUX are not much worse than for NEW, except for the stations HKO, HKP, and TY1, which are surrounded by high-rise buildings. Results for HW2009 mainly corroborate

those for HW2018. In contrast to HW2018, particularly bad model performance is also found for station HPV, which is located to the north-west of a high-rise high-density district. The model results for CLASSICAL may therefore be worsened due to the easterly wind direction. Model results are also bad for the stations on Kowloon Peninsula (KLT and KP). However, the result at KP for D4 is in noted contrast to the result for D5, where at higher model resolution, the environment of the station is better represented. NEW leads to better model results for all urban stations, except for CPH, which is located on the roof of a 62 m

tall building and therefore does not suffer from the issues with the SBL scheme as the stations which measure near the surface. Even for NEW, a positive temperature bias of about 1 K prevails for the urban and rural stations, which is most probably due to the overestimation of the total downwelling solar radiation and to a lesser degree by the fact that too many buildings are in the model domain since the dataset on urban form and function represents the 2018 situation and the population of Hong Kong has increased by about 1 million since 2009. Air temperature in rural areas is also generally better simulated for NEW during

both heat waves, although the improvement is not as marked as for the urban stations.

Evaluation results for relative humidity are consistent with those for air temperature. The urban stations that exhibit the largest positive bias for air temperature exhibit the largest negative bias for relative humidity. For CLASSICAL, unacceptably high values for the rmse are found for the stations HKO, HKS, JKB, TWN, and TY1 for HW2018. These stations also exhibit unacceptable results for air temperature. NEW improves the bias and rmse for all urban stations, but negative biases of around

5% are still found, which is consistent with the positive bias of 0 to 1 K for air temperature. For HW2009, NEW results in improved bias and rmse for all urban stations, but even for NEW, the positive temperature bias leads to a negative bias of relative humidity between 5 and 10%. Evaluation measures for relative humidity are also improved in the rural areas, but not as much as in the urban areas. SURFFLUX differs in a relevant manner from NEW only at the stations HKO, TWN, and TY1, which are surrounded by high-rise buildings.

CLASSICAL leads to a positive wind speed bias in all urban stations, except station HKO for both heat waves and station HKS for HW2009. NEW leads to lower values of the average wind speed for all urban stations, except CP1 for HW2009. This improves most of the stations with a positive bias, but deteriorates results for the station HKO. The rmse of wind speed at most urban stations is considerably reduced for both heat waves, especially at the stations SHA and SEK, which measure wind speed in 10 m a.g.l. and therefore have their simulated values taken from the SBL scheme of TEB in CLASSICAL. At the station KP,

where wind is measured in 25 m a.g.l. NEW reduces the wind speed too much compared to CLASSICAL, maybe because the drag force due to the buildings alters too strongly the wind speed of the station located in an urban park. The rmse for station KP is improved for HW2018, but slightly deteriorated for HW2009. Wind speed at stations SHA and SEK is also considerably improved for NEW and HW2009. At the rural stations, evaluation measures for wind speed are slightly improved for HW2018 and not consistently changed for HW2009.

NEW does not improve the evaluation measures for wind speed as much as for air temperature and relative humidity. This is due to the way the wind speed is diagnosed for CLASSICAL (Figure 1). The wind speed values from the TEB SBL levels are





taken if the height of the wind anemometer is below the height of the highest SBL level. With this approach to diagnose the wind speed values, the lack of friction due to the high-rise buildings in the atmospheric model does not influence the model evaluation measures too much. However, this does not change the fact that the high-rise buildings do not directly influence

multiple atmospheric model levels. In order to illustrate this, the fields of wind and air temperature at 30 m a.g.l. simulated by Meso-NH in the daytime (11 to 16 Local Time) during HW2009 and HW2018 are displayed in Figure 10. For HW2009, Kowloon Peninsula is ventilated by a sea breeze from the south-east, which for CLASSICAL (Figure 10a) is not sufficiently influenced by the high-rise buildings on Kowloon Peninsula. For NEW (Figure 10b), a deflection of the sea breeze at the east of Kowloon Peninsula towards the Kai Tak area is simulated, which is not found for CLASSICAL. Furthermore, the wind

direction is south to south-west on the western coast of Kowloon Peninsula for NEW, whereas it is mainly south-east for CLASSICAL. For CLASSICAL, the air penetrates easily the areas with very high buildings along the northern coast of Hong Kong Island, whereas the wind speed is considerably reduced in this region for NEW. For HW2018, the south-westerly sea-breeze appears to penetrate too efficiently the Kowloon Peninsula for CLASSICAL (Figure 10c), whereas it is considerably slowed down by the high-rise buildings there for NEW (Figure 10d). Although all these features cannot be validated by field

observations in the present study, they appear physically more plausible for NEW than for CLASSICAL.

For wind direction (not shown), the rmse increases for the NEW coupling approach at nearly all stations. This could be due to enhanced spatio-temporal wind direction fluctuations due to the spatially heterogeneous drag force, and as a result a worse agreement with the observations than for a more homogeneous wind field.

## 4.2 Vertical profiles of meteorological variables

The vertical profiles of air temperature and wind speed are evaluated with the radiosoundings made at King's Park. Since temporal averaging of the vertical profiles is not useful, the results are displayed as an example for May 20 2018 (Figure 11), a day with westerly wind (Figure 5), leading to advection of air from the dense mid- to high-rise setting west of the station. The vertical profiles are extracted only from the location of the station, neglecting the displacement of the radiosonde with the wind. This is a reasonable assumption since only the lowest 200 m are investigated.

At 6 UTC (14 Local Time), CLASSICAL strongly overestimates the air temperature at the grid points closest to the surface for which the SBL scheme is employed. The deviation from the radiosounding increases as it gets closer to the surface. The vertical profiles agree much better with the radiosounding for NEW and SURFFLUX. These results are consistent with those for the time series of near-surface air temperature for station KP (Figure 5). At 12 UTC (20 Local Time), which is after sunset in Hong Kong, the radiosounding indicates that the atmosphere has become stable at the lowest 40 m a.g.l.. This is under-

standable, since the radiosounding is made inside a small park. The model is not able to capture this, which is most likely because the model grid point is not free of buildings. The positive sensible heat flux from the building walls and roof to the air keeps the atmosphere unstable. Very high-resolution simulations would be needed to capture the environment of an urban oasis like KP correctly. The results for SURFFLUX are worse than for NEW, since for SURFFLUX the sensible heat flux from the walls and the roofs is coupled at the surface of the atmospheric model. For CLASSICAL, the overestimation of near-surface

air temperature is even larger, maybe because the buildings or the ground have stored more heat during the day due to the





strong overestimation of air temperature in the daytime. As a result, NEW improves the results even though it does not capture the stable layer below 40 m a.g.l.. In the nighttime (18 UTC; 2 Local Time), the stable layer extends up to 60 m a.g.l. with a marked inversion below. Similar to the results for 12 UTC, this cannot be captured by NEW, although it performs better than SURFFLUX and CLASSICAL. The vertical profiles of air temperature have been analysed for other days and relatively

similar results are found (not shown). For 0 UTC (8 Local Time, not shown) there is only little difference for air temperature for the different coupling approaches. CLASSICAL exhibits the same tendency to overestimate air temperature very close to the surface, but the difference from NEW is much lower than for 6 UTC.

For wind speed, much larger discrepancies between the simulation results and the radiosoundings are found than for air temperature. This can be due to shortcomings of the model or the lack of spatial representativeness of the radiosounding compared

to the grid point scale model result, but also due to the turbulent fluctuations of the wind in this very heterogeneous urban environment. CLASSICAL overestimates the wind speed for 6, 12, and 18 UTC, most probably since there is not sufficient building drag. For NEW and SURFFLUX, the drag force leads to lower wind speed values, and as a result a better agreement with the radiosounding. However, the shapes of the profiles do not agree very well and visual inspection for other days reveals sometimes different results, e.g. CLASSICAL matches the observed profile better than NEW and SURFFLUX for some ra-

diosoundings. For HW2018 with 60 available radiosoundings, CLASSICAL overestimates the wind speed in between the top of the SBL and 150 m a.g.l. for 44 out of 60 radiosoundings. The wind speed profile agrees better for NEW than for CLASSICAL for 41 out of 60 radiosoundings. For HW2009, CLASSICAL overestimates the wind speed in between the top of the SBL and 150 m a.g.l. for 11 out of 16 radiosoundings and a better agreement is found for NEW for 9 out of the 16 radiosoundings. For HW2018, with a wind from the high-rise districts west of the station, the wind profiles are more frequently improved than

for HW2009 with a wind from more open areas east of the station.





**Figure 8.** Model evaluation measures for hourly time series at meteorological stations in D4 and for HW2018. The urban stations are bold, the stations on mountain peaks are marked with a *.



**Figure 9.** Same as Figure 8, but for HW2009.



(a) CLASSICAL, Sep. 7-8 2009

(b) NEW, Sep. 7-8 2009

(c) CLASSICAL, May 28-30 2018

(d) NEW, May 28-30 2018

**Figure 10.** Wind field and air temperature at 30 m a.g.l. in the daytime (11 to 16 Local Time) for the CLASSICAL and NEW coupling approaches and two selected time periods during HW2018 and HW2009.







**Figure 11.** Vertical profiles of air temperature and wind speed at station King's Park. H-SBL indicates the height below which for CLASSI-CAL the meteorological variables are taken from the SBL scheme.





**Table 2.** Average anthropogenic heat flux due to building energy consumption ($\overline{Q_A^{bld}}$) and total building energy consumption ($E_A^{bld}$).

| Domain | $\overline{Q_A^{bld}}$ [Wm$^{-2}$] | | $E_A^{bld}$ [TJ] | |
|---|---|---|---|---|
| | Model | Inventory | Model | Inventory |
| D4 | 22.0 | 19.9 | 1464 | 1324 |
| D5 | 49.6 | 43.8 | 1169 | 1031 |

### 4.3 Anthropogenic heat flux due to buildings

For the NEW coupling approach, the magnitude and spatial distribution of the monthly average anthropogenic heat flux due to buildings is evaluated for May 2018. It is assumed that buildings are the only contributors to the city's energy consumption in the domestic, industrial, and commercial sectors. Overall, there is a slight overestimation of the monthly average anthropogenic heat flux for D4 and D5 of around 11% and 13%, respectively (Table 2). Otherwise, there is generally a good agreement in the spatial distribution between the model and the inventory (Figure 12). The central business district along the northern coast of Hong Kong Island sees the highest anthropogenic heat flux of up to above 500 Wm$^{-2}$. Commercial and industrial areas around Tsim Sha Tsui (southern tip of Kowloon Peninsula) and Kwun Tong (east Kowloon) also have a high anthropogenic heat flux between 100 and 500 Wm$^{-2}$. Other highly urbanised areas in the Kowloon Peninsula, north-eastern and western coasts of Hong Kong Island, as well as the Shatin, Tsuen Wan, and Tseung Kwan O new towns exhibit relatively lower values (between 25 and 100 Wm$^{-2}$).

The modelled anthropogenic heat flux is further evaluated for each building archetype (Figure 13). The model is able to capture the different magnitudes of heat fluxes for different building type and functions, but there is a considerable overestimation at grid points with the dominant building type of hotel, industrial building, and hospital. According to the authors' local knowledge, the fact that many industrial buildings in Hong Kong have been converted into storage warehouses, private workshops, retail shops, restaurants etc., which do not use as much energy nor follow the same behavioural schedules of the assumed industrial activities, may be the reason for this overestimation. As for hotels and hospitals, the building energy consumption is high owing to their 24/7 occupancy. However, the overestimation is likely caused by the grid-dominant building type approach, as not all buildings within the same model grid belong to such energy-intensive uses. The overestimation for the three private housing building types (Private Housing, Newer Private Housing, Modern Private Housing) may be attributable to too high internal heat loads and the assumed nighttime air conditioning, which may not be representative for all occupants with different financial ability, environmental awareness, and thermal comfort acceptability. On the contrary, there is an underestimation of anthropogenic heat flux for commercial buildings (Commercial Skyscraper, Old Commercial Building), probably because of uncertainties in the behaviour and occupancy settings of the buildings with office uses, as it is difficult to obtain real surveyed data for these buildings due to privacy or security issues. The underestimation of energy consumption for buildings of transport-related uses and historical monuments may be explained by the missing mechanical ventilation in the model for these building types and the mix of neighbouring buildings with other uses in the same grid not taken in account by the grid-dominant



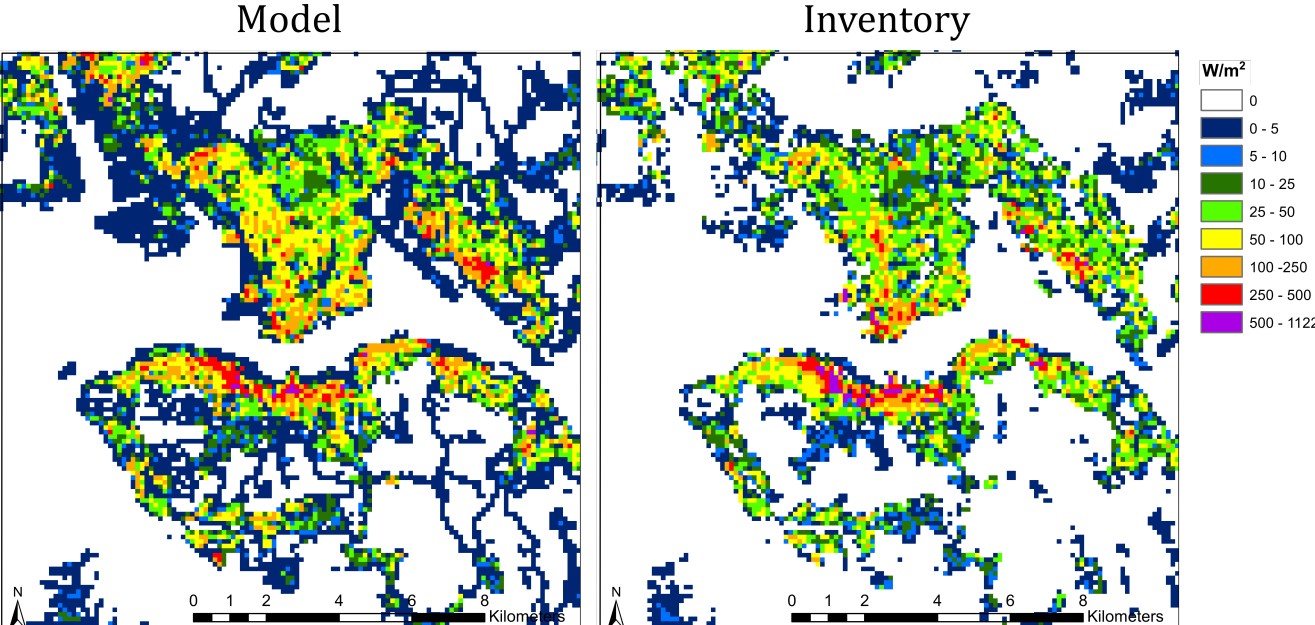

**Figure 12.** Anthropogenic heat flux due to building energy consumption in D5.

building type approach. Despite the discussed uncertainties of the model and those of the inventory (Section 3.3.2), the results are encouraging and confirm the applicability of the TEB coupled to BEM in a city with complex and heterogeneous urban

form and function.

## 5 Discussion

### 5.1 Comparison with previous studies

Model evaluation reveals a marked improvement of the simulated near-surface air temperature and relative humidity for the new multi-layer coupling between SURFEX-TEB and Meso-NH compared to the previous single-layer coupling. The average

values of the bias and rmse for air temperature and relative humidity at urban and rural stations obtained for the present and previous studies are presented in Tables 3 and 4. In the present study, stations are considered as urban if in a circle with 250 m radius centered at the station the plan area building density is above 0.1 or the average building height is above 15 m. All other stations are considered as rural even though they are characterised by a large variety of environments like forests, small islands and mountain peaks. The precise definition of the urban and rural stations is not given in all previous studies,

therefore a certain degree of uncertainty remains when comparing the model evaluation measures with previous studies. For the CLASSICAL single-layer coupling, the model results are not of acceptable quality. For the NEW multi-layer coupling, the rmse of air temperature obtained for both heat waves is very similar to the values obtained for the previous WRF-BEP applications in Hong Kong. Model improvement is the largest at urban stations, which is no surprise since the most relevant changes of the



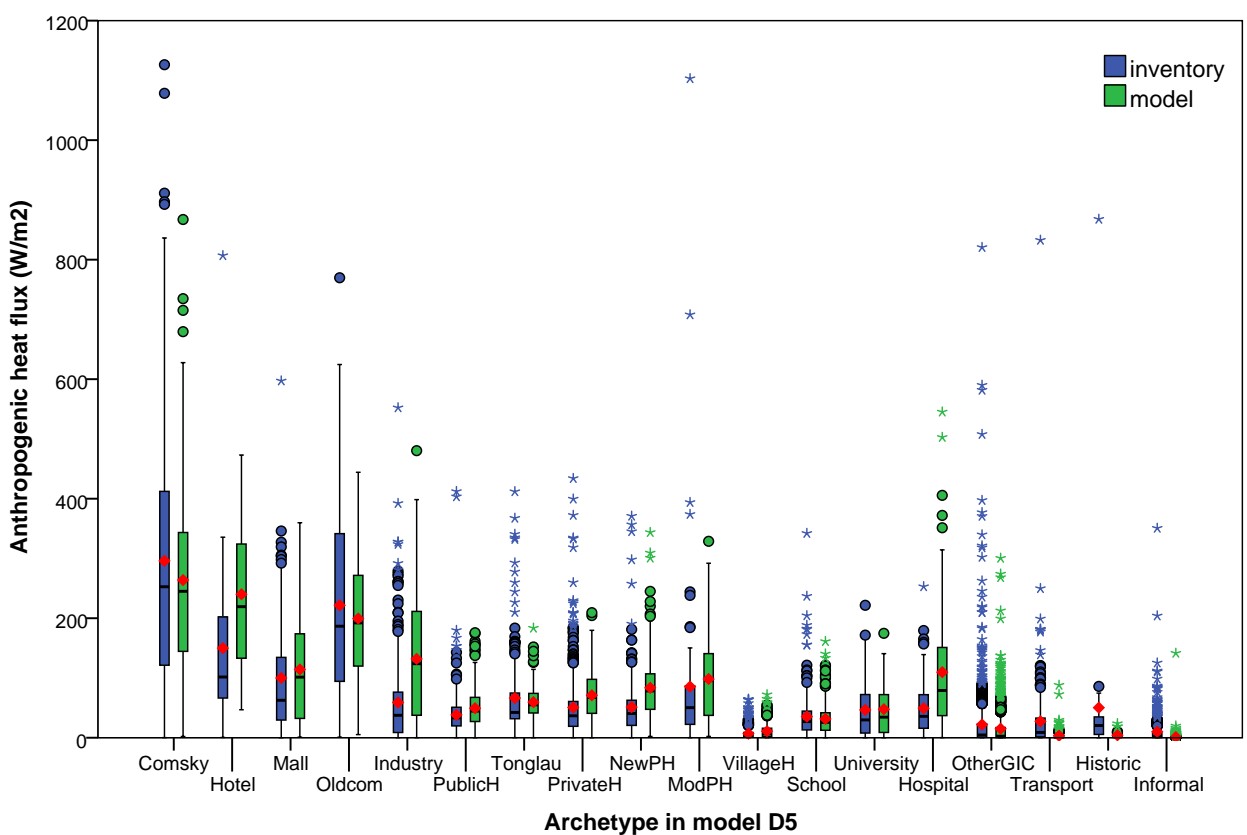

**Figure 13.** Statistical distribution of the anthropogenic heat flux due to building energy consumption in D5 as a function of the dominant building archetype at grid point scale. In the boxplots, the black line denotes the median and the red diamond the mean. Comsky denotes Commercial Skyscraper, Oldcom Old Commercial Building, PublicH (PrivateH) Public (Private) Housing, NewPH (ModPH) Newer (Modern) Private Housing, VillageH Village House, other GIC other Government, Institutional, and Community buildings, and Historic Historical Building.

coupling concern the urban environment. Interestingly, the lowest rmse of 1.0 K among all previous studies is reported by Lo
et al. (2007) for a relatively coarse model resolution of 1.5 km and the simple urban parametrisation in the Noah LSM. This good performance might be due to the low values of solar radiation, and hence the low daily temperature amplitude, in their short simulation period at the end of October. The present study overestimates near-surface air temperature and underestimates near-surface relative humidity. Despite this, the rmse of relative humidity for the NEW coupling approach indicates similar or slightly higher model quality compared to previous studies. Another interesting finding is that coupling the heat and moisture
fluxes from the building walls and roofs at the surface deteriorates the simulation results only for those stations surrounded by buildings of at least 40 m. It might therefore be neglected during model applications at a coarser resolution.

Compared with previous studies, the Meso-NH-TEB results for wind speed are of similar to better quality for both the single





**Table 3.** Comparison of evaluation measures for air temperature with previous studies. $\Delta$ is the model resolution.

|  |  |  |  |  | bias [K] | | | rmse [K] | | |
| --- | --- | --- | --- | --- | --- | --- | --- | --- | --- | --- |
| Study | Period | Model | $\Delta$ [m] | Obs. | urb. | rur. | all | urb. | rur. | all |
| Present | 1-8/9/2009 | MNH-TEB-CLASSICAL | 250 | 24 HKO | 2.1 | 1.5 | 1.8 | 2.6 | 2.0 | 2.4 |
| Present | 1-8/9/2009 | MNH-TEB-NEW | 250 | 24 HKO | 0.9 | 0.8 | 0.8 | 1.4 | 1.4 | 1.4 |
| Present | 17-31/5/2018 | MNH-TEB-CLASSICAL | 250 | 29 HKO | 1.8 | 1.4 | 1.6 | 2.2 | 1.9 | 2.1 |
| Present | 17-31/5/2018 | MNH-TEB-NEW | 250 | 29 HKO | 0.7 | 0.7 | 0.7 | 1.2 | 1.6 | 1.4 |
| Wong et al. (2019) | 18-22/12/2010 | WRF-NoahLSM | 500 | 12 HKO | -0.4 | - | - | 1.5 | - | - |
| Wong et al. (2019) | 18-22/12/2010 | WRF-BEP-BEM | 500 | 12 HKO | -0.0 | - | - | 1.4 | - | - |
| Wang et al. (2018) | 23-28/6/2016 | WRF-BEP-BEM | 500 | 27 HKO | -0.1 | - | 0.2 | - | - | 1.2 |
| Wang et al. (2017) | 15-18/9/2012 | WRF-BEP-BEM | 500 | 25 HKO | - | - | 0.1 | - | - | 1.4 |
| Wang et al. (2017) | 15-18/9/2012 | WRF-BEP-BEM (new $c_d$) | 500 | 25 HKO | - | - | 0.3 | - | - | 1.6 |
| Wang et al. (2014) | entire 2008 | WRF-SLUCM | 4000 | 5 PRD | - | - | 0.7 | - | - | 1.4 |
| Lo et al. (2007) | 30-31/10/2003 | MM5-bulk | 1500 | 33 HKO | -0.3 | -0.2 | -0.3 | 2.0 | 1.2 | 1.7 |
| Lo et al. (2007) | 30-31/10/2003 | MM5-NoahLSM | 1500 | 33 HKO | -0.2 | -0.3 | -0.2 | 1.0 | 1.0 | 1.0 |
| Lam et al. (2006) | 28/12/99-1/1/00 | MM5-NoahLSM | 1500 | 7 HKO | - | - | - | - | - | 1.7 |

**Table 4.** Same as Table 3, but for relative humidity.

|  |  |  |  |  | bias [%] | | | rmse [%] | | |
| --- | --- | --- | --- | --- | --- | --- | --- | --- | --- | --- |
| Study | Period | Model | $\Delta$ [m] | Obs. | urb. | rur. | all | urb. | rur. | all |
| Present | 1-8/9/2009 | MNH-TEB-CLASSICAL | 250 | 19 HKO | -11.3 | -9.8 | -10.5 | 13.1 | 12.4 | 12.8 |
| Present | 1-8/9/2009 | MNH-TEB-NEW | 250 | 19 HKO | -7.7 | -6.4 | -7.0 | 10.1 | 10.4 | 10.3 |
| Present | 17-31/5/2018 | MNH-TEB-CLASSICAL | 250 | 16 HKO | -9.5 | -8.7 | -9.1 | 11.0 | 11.7 | 11.3 |
| Present | 17-31/5/2018 | MNH-TEB-NEW | 250 | 16 HKO | -5.3 | -5.3 | -5.3 | 7.6 | 10.5 | 8.9 |
| Wang et al. (2017) | 15-18/9/2012 | WRF-BEP-BEM | 500 | 15 HKO | - | - | 3.1 | - | - | 10.2 |
| Wang et al. (2017) | 15-18/9/2012 | WRF-BEP-BEM (new $c_d$) | 500 | 15 HKO | - | - | 2.9 | - | - | 11.0 |
| Wang et al. (2014) | entire 2008 | WRF-SLUCM | 4000 | 5 PRD | - | - | -5.2 | - | - | 13.1 |

and multi-layer coupling. The good results for the single-layer coupling are only obtained because the simulated wind speed values from the SBL levels below the physical surface are taken. The multi-layer coupling is therefore beneficial for the simple reason that the influence of the buildings on the wind speed in the mesoscale model is better represented.





**Table 5.** Same as Table 3, but for wind speed.

| Study | Period | Model | Δ [m] | Obs. | bias [ms$^{-1}$] | | | rmse [ms$^{-1}$] | | |
|---|---|---|---|---|---|---|---|---|---|---|
| | | | | | urb. | rur. | all | urb. | rur. | all |
| Present | 1-8/9/2009 | MNH-TEB-CLASSICAL | 250 | 16 HKO | 0.4 | -0.6 | -0.2 | 1.4 | 2.0 | 1.8 |
| Present | 1-8/9/2009 | MNH-TEB-NEW | 250 | 16 HKO | -0.2 | -1.4 | -0.8 | 1.3 | 2.2 | 1.8 |
| Present | 17-31/5/2018 | MNH-TEB-CLASSICAL | 250 | 18 HKO | 1.0 | 1.2 | 1.1 | 1.5 | 1.9 | 1.8 |
| Present | 17-31/5/2018 | MNH-TEB-NEW | 250 | 18 HKO | 0.3 | 1.0 | 0.7 | 1.2 | 1.8 | 1.5 |
| Dy et al. (2019) | 1-31/7/2010 | WRF-basic-ACM | 1000 | 352 PRD | 0.6 | 0.7 | 0.7 | 2.2 | 2.4 | 2.3 |
| Dy et al. (2019) | 1-31/7/2010 | WRF-urban-ACM | 1000 | 352 PRD | -0.2 | 0.5 | 0.1 | 1.7 | 2.3 | 2.0 |
| Dy et al. (2019) | 1-31/12/2010 | WRF-basic-ACM | 1000 | 352 PRD | 1.0 | 1.2 | 1.1 | 2.5 | 3.0 | 2.7 |
| Dy et al. (2019) | 1-31/12/2010 | WRF-urban-ACM | 1000 | 352 PRD | 0.0 | 0.7 | 0.5 | 1.6 | 2.3 | 1.9 |
| Wong et al. (2019) | 18-22/12/2010 | WRF-NoahLSM | 500 | 12 HKO | 0.6 | - | - | 1.6 | - | - |
| Wong et al. (2019) | 18-22/12/2010 | WRF-BEP-BEM | 500 | 12 HKO | -0.2 | - | - | 1.0 | - | - |
| Wang et al. (2017) | 15-18/9/2012 | WRF-BEP-BEM | 500 | 18 HKO | - | - | 0.7 | - | - | 2.2 |
| Wang et al. (2017) | 15-18/9/2012 | WRF-BEP-BEM (new $c_d$) | 500 | 18 HKO | - | - | 0.2 | - | - | 2.2 |
| Wang et al. (2014) | entire 2008 | WRF-SLUCM | 4000 | 5 PRD | - | - | 0.5 | - | - | 1.3 |
| Lam et al. (2006) | 28/12/99-1/1/00 | MM5-NoahLSM | 1500 | 7 HKO | - | - | - | - | - | 1.7 |

## 5.2 The relevance of horizontal advection for near-surface air temperature

The benefit of the NEW coupling approach is that it allows one to take into account the horizontal advection in the urban canopy layer, e.g. from the cooler sea or forests into the warmer high-rise high-density urban environment. Theoretically, the more heterogeneous the land use and urban morphology and the larger the horizontal meteorological variable gradients, the larger the benefit of considering horizontal advection. To quantify the contribution of horizontal advection, average daily cycles of the different terms in the prognostic equation for potential temperature in Meso-NH are calculated for the entire HW2009 and HW2018 in the lowest 30 m of the atmosphere and the most relevant terms are displayed in Figure 14 for two boxes covering Kowloon Peninsula and the high-rise district in the north-west of Hong Kong Island (Figure 10b). The results shown are for the NEW coupling approach. For the CLASSICAL coupling approach, there is no advection in the urban canopy layer, so the advection term is 0 by definition.

For both heat waves and both boxes, the temporal evolution of the near-surface potential temperature is mainly governed by the warming due to the sensible heat fluxes from the buildings, and the cooling due to horizontal advection and vertical diffusion. On Kowloon Peninsula, the building heat fluxes are larger in the daytime since building walls and roofs are heated by solar radiation and release a large part of the heat immediately. However, this is not the case in the high-rise district in the north-west of Hong Kong Island. The daily cycle of the building heat fluxes is less marked there, probably since in this high-rise district more heat is stored in the building materials during the day and released during the night. For Kowloon Peninsula, the





advection contributes to reducing the near-surface air temperature in the same order of magnitude than the vertical diffusion during the daytime and in the evening. Horizontal advection is only of low importance in the nighttime. For the north-west of Hong Kong Island, the horizontal advection is of high importance during the entire day and both heat waves, which is due to

the vicinity to the coastline with channelling of the wind between Hong Kong Island and Kowloon Peninsula. The results of the budget analysis corroborate the finding that the NEW coupling approach leads to a reduction of near-surface air temperature and therefore to a better agreement with the HKO observations. This is at least partly due to the consideration of horizontal advection in the very heterogeneous environment of Hong Kong. However, vertical diffusion is also important, and therefore model results will likely also be influenced by the choice of the turbulent mixing length. A modification of the mixing length

in the SBL scheme to enhance vertical mixing might lead to better results for the single-layer coupling, but not necessarily for the right reason.

## 5.3   Drag force approach and urban turbulent length scales

The NEW coupling approach using a drag coefficient for the walls of $C_d = 0.4$ leads to an underestimation of the wind speed values at the stations in the urban parks. This is in contrast to Santiago and Martilli (2010) who found an overestimation

for wind speed in the urban canopy with the same drag force approach and the same value of $C_d = 0.4$ when compared to obstacle-resolving Computational Fluid Dynamic results. This discrepancy might be due to the fact that the HKO stations are located in small urban parks, and therefore their environment is not sufficiently resolved. The fact that there are buildings, and subsequently a drag force due to the walls and roofs applied at the station grid point might explain the underestimation of the wind speed at the urban stations. The results of the present study are also in contrast to Gutiérrez et al. (2015), who found

an overestimation of wind speed for a WRF-BEP application to New York. However, this might be due to the use of rooftop stations for model evaluation by Gutiérrez et al. (2015). More observations of wind speed from inside the urban canopy layer of high-rise high-density cities are required to be able to judge whether the formulation of the drag force approach or the value of the drag coefficient has to be improved.

No modification of the length scales for turbulent mixing and dissipation has been made. Tests have been performed using

the turbulent length scales proposed by Santiago and Martilli (2010) for the Meso-NH model levels lower than two times the average building height, but not more than 40 m a.g.l. This deteriorated model results for air temperature and relative humidity, leading to too high (low) air temperature (relative humidity) when compared to the meteorological stations. This indicates that the vertical turbulent exchange is underestimated when using the lower values of the turbulent length scales specific to the urban environment. However, this deficiency might be due to the fact that the length scales defined by Santiago and Martilli

(2010) have been derived for very idealised urban morphologies and neutral conditions. Further studies are needed to derive urban turbulent length scales for more realistic urban morphologies and a variety of atmospheric stability regimes.

## 5.4   The relevance of the sea surface temperature

The SST values are very important for the correct simulation of the meteorological conditions in Hong Kong. For HW2018, the authors noted a very strong increase (up to 4.5 K) of the SST between the beginning and the end of the heat wave reaching



**Figure 14.** Daily cycle of the most relevant terms in the prognostic equation for the potential temperature for the NEW coupling approach in the lowest 30 m of the atmosphere for the two boxes covering Kowloon Peninsula and the north-west of Hong Kong Island displayed in Figure 10b.

values that are 2 K higher than the point observation in the harbour. Therefore, one reason for the positive bias at the end of HW2018 might be the too high SST. A possible source of uncertainty for the SST in Hong Kong is a wrong estimation of the temperature and volume of the freshwater from the Pearl river in the SST analysis. Another source of uncertainty is the coarse horizontal resolution of the SST analysis of about 10 km, which might not sufficiently resolve cold upwelling close to





the shore. In a future study, a coupled atmosphere-ocean model might be applied to dynamically simulate the state of the sea
at high resolution as a function of the meteorological conditions and the freshwater influx from the Pearl river.

## 5.5   Anthropogenic heat flux

The monthly average values of the anthropogenic heat flux due to the buildings for the city of Hong Kong are above 500
$\mathrm{Wm}^{-2}$ in the high-rise high-density districts. These values are of similar magnitude than the solar radiation which is usually
the main driver of the Earth's surface energy balance. Since similar values can be expected for other, more extensive Asian
megacities, it might be worth representing these heat fluxes in the new generation of very high resolution Earth system models.
Meso-NH coupled with SURFEX-TEB-BEM is able to simulate the monthly average building-related anthropogenic heat flux
with an overestimation of about 10%, which could be due to the positive temperature bias of 0 to 1 K at the urban stations for
the simulation covering entirely May 2018. This is remarkable given the large number of uncertain input parameters related
to urban morphology, building construction materials, capacity and coefficient of performance of air conditioning systems,
building use, and occupant's behaviour (Masson et al., 2020).

## 6   Conclusions

In the present study, the multi-layer coupling of the urban canopy model Town Energy Balance (TEB) included in the land
surface model SURFEX with the mesoscale atmospheric model Meso-NH has been introduced. The main objective of the
new multi-layer coupling is to better represent the interactions between high-rise cities and the atmosphere. This is a step
towards future high-resolution weather prediction models with a horizontal resolution of about 100 m and studies quantifying
the impact of climate change mitigation and adaptation measures implemented in high-rise high-density cities. Such high-rise
settings are very common in the young Asian megacities and are becoming more prevalent in newly constructed urban districts
in other parts of the world.

The introduced multi-layer coupling is simple. The geometric assumption in TEB that all buildings at grid point scale have the
same height and are aligned along a street canyon of infinite length to calculate the radiative exchanges in the urban canopy
layer is unchanged. To maintain the coherence between the calculations in TEB and Meso-NH, the effect of the buildings on
the atmosphere is only considered up to the average building height. The effect of the buildings on the prognostic variables of
Meso-NH is taken into account using a drag force approach which reduces the horizontal wind components representing the
friction due to the building walls and roofs and increases the turbulent kinetic energy representing the production of kinetic
energy due to the wind shear close to the buildings. The heat and moisture fluxes from the building walls and roofs are released
at the atmospheric model levels intersecting these urban facets. No modifications of the length scales for turbulent transport
and dissipation have been made in the present study.

The multi- and single-layer coupling approaches have been tested for two selected prolonged heat waves in the heterogeneous
high-rise high-density city of Hong Kong, since for this city high-quality data on urban form and function as well as a dense
network of meteorological stations are available. With the single-layer coupling, model results for near-surface air temperature





and relative humidity are of poor quality, which is expected since the single-layer version of TEB was not initially developed for high-rise heterogeneous cities. The new multi-layer coupling leads to a strong improvement of the model results, bringing the model performance on par with, if not better than, the previous applications with the more complex multi-layer WRF-BEP model in Hong Kong. Evaluation of the vertical profiles in the lower boundary layer with radiosounde observations indicates

that for the single-layer coupling approach, the deviation from the observation mainly occurs in the urban canopy layer where the 1D Surface Boundary Layer scheme is employed to calculate vertical profiles of the meteorological variables. This is due to the lack of the consideration of horizontal advection of air temperature from the cooler surrounding rural areas or the sea towards the warmer urban environment. For the wind speed, the model results are improved on average for the multi-layer coupling approach, but not for all stations and all situations. The effect of the buildings on the Meso-NH model levels is clearly

underestimated with the single-layer coupling approach and this leads to considerable differences in small scale circulation features.

The most important future enhancement of the multi-layer SURFEX-TEB will be the modification of the radiative exchange calculations using recent developments of Hogan (2019a) and Hogan (2019b). With these, it will be possible to consider a variety of building heights at grid point scale and as a consequence also for the drag force, heat and moisture fluxes in Meso-

NH. This should improve the model results in areas not conforming to the urban canopy assumption (e.g. building clusters standing atop podiums) or areas with isolated high-rise buildings in otherwise low- to mid-rise settings. Such situations are not well represented in the current multi-layer coupling. The improved treatment of urban radiation can also allow one to take into account the effect of urban air pollution or urban fog, which will become more relevant as the number of high-rise buildings in a city increases.

The evaluation of the new multi-layer coupling has suffered from the lack of observations that are actually representative of the urban canopy layer, since in Hong Kong, even the most urban stations are actually located in small parks. Therefore, it is very difficult to judge based on the presented model evaluation whether the choices for the drag coefficient, or the turbulent length scales are actually justified. Further observation campaigns in high-rise high-density cities should therefore focus on obtaining more observations of meteorological parameters from inside the urban canopy layer.

Further work is required to derive, test, and evaluate the different drag force approaches and urban turbulent length scales. It needs to be determined whether it is worthwhile to also take into account the directional variations of the drag coefficient due to the building shape or urban morphology. This could represent processes like channelling in the streets, variations with atmospheric stability, or even a breakdown of the underlying theoretical framework for high-density cities since there is too much sheltering. Obstacle-resolving modelling for a large variety of idealised urban morphologies and meteorological

situations needs to be employed to derive more robust formulations for the drag coefficients and the turbulent length scales.

*Code and data availability.* The modified source code containing routines of SURFEX-TEB and Meso-NH is provided in the Supplement. A short documentation explains in which routines the equations presented in the manuscript can be found. Furthermore, the simulation





directories for the different coupling approaches and the two heat waves are provided. Further information is available on request. Model developments will be merged with the official version of SURFEX-V9.0 and Meso-NH-V5.4.

**Appendix A: Station meta data**

The meta data for the meteorological station network operated by the Hong Kong Observatory are given in Table A1.

**Appendix B: Model evaluation measures for highest resolution (125 m) domain (D5)**

Model evaluation measures for air temperature, relative humidity, and wind speed for the highest resolution (125 m) domain D5 are given in Figure B1 (B2) for HW2018 (HW2009).





**Table A1.** Meta data concerning the station network and land cover in their surrounding. $\lambda_p$ and $\lambda_i$ denote the plan area building and impervious fraction respectively. The height a.g.l. of the Stevenson screen is about 1 m, except for station CPH for which it is 62 m.

| Station code | Latitude [°] | Longitude [°] | Ground elevation [m] | Height of wind anemometer [m] | $\lambda_p$ (125 m) | $\lambda_i$ (125 m) | $H_{bld}$ [m] (125 m) | $\lambda_p$ (250 m) | $\lambda_i$ (250 m) | $H_{bld}$ [m] (250 m) |
|---|---|---|---|---|---|---|---|---|---|---|
| BHD | 22.1975 | 114.2119 | 94 | 9 | - | - | - | 0.00 | 0 | 4.5 |
| CCH | 22.2011 | 114.0267 | 72 | 27 | - | - | - | 0.02 | 0.02 | 13.2 |
| CP1 | 22.2889 | 114.1558 | 3 | 27 | 0.14 | 0.0 | 12.3 | 0.10 | 0.08 | 10.0 |
| CPH | 22.3481 | 114.1092 | 61 | - | 0.17 | 0.48 | 61.1 | 0.09 | 0.35 | 54.3 |
| GI | 22.2850 | 114.1128 | 88 | 19 | 0.03 | 0.0 | 7.2 | 0.01 | 0.00 | 7.4 |
| HKO | 22.3019 | 114.1742 | 32 | 42 | 0.26 | 0.20 | 29.3 | 0.39 | 0.30 | 40.6 |
| HKP | 22.2783 | 114.1622 | 26 | - | 0.12 | 0.07 | 10.2 | 0.16 | 0.25 | 93.4 |
| HKS | 22.2478 | 114.1736 | 5 | 25 | 0.00 | 0.46 | 3.8 | 0.17 | 0.37 | 16.8 |
| HPV | 22.2706 | 114.1836 | 5 | - | 0.01 | 0.05 | 3.0 | 0.07 | 0.21 | 32.7 |
| JKB | 22.3158 | 114.2556 | 38 | 14 | - | - | - | 0.11 | 0.10 | 13.5 |
| KFB | 22.4328 | 114.1208 | 307 | - | - | - | - | 0.01 | 0.12 | 4.5 |
| KLT | 22.3350 | 114.1847 | 92 | - | 0.01 | 0.03 | 3.6 | 0.03 | 0.05 | 15.3 |
| KP | 22.3119 | 114.1728 | 65 | 25 | 0.05 | 0.07 | 4.0 | 0.04 | 0.05 | 16.8 |
| KSC | 22.3703 | 114.3125 | 39 | - | - | - | - | 0.0 | 0.0 | 0.0 |
| KTG | 22.3186 | 114.2247 | 90 | - | 0.05 | 0.04 | 3.2 | 0.07 | 0.10 | 35.3 |
| LAM | 22.2261 | 114.1086 | 7 | 10 | 0.11 | 0.01 | 7.5 | 0.03 | 0.00 | 7.5 |
| PEN | 22.2911 | 114.0433 | 34 | 13 | - | - | - | 0.02 | 0.00 | 4.2 |
| SE | 22.3097 | 114.2133 | 1 | 15 | 0.03 | 0.00 | 6.2 | 0.01 | 0.05 | 6.2 |
| SE1 | 22.3047 | 114.2172 | 4 | - | 0.01 | 0.11 | 5.0 | 0.00 | 0.05 | 5.0 |
| SEK | 22.4361 | 114.0847 | 16 | 10 | - | - | - | 0.16 | 0.27 | 7.3 |
| SHA | 22.4025 | 114.21 | 6 | 10 | - | - | - | 0.13 | 0.47 | 10.8 |
| SKG | 22.3756 | 114.2744 | 4 | 28 | - | - | - | 0.09 | 0.32 | 8.32 |
| SKW | 22.2817 | 114.2361 | 53 | - | 0.02 | 0.12 | 8.1 | 0.02 | 0.08 | 5.9 |
| SSP | 22.3358 | 114.1369 | 11 | - | 0.21 | 0.09 | 51.6 | 0.20 | 0.17 | 48.9 |
| STY | 22.2142 | 114.2186 | 31 | - | - | - | - | 0.18 | 0.33 | 11.9 |
| TC | 22.3578 | 114.2178 | 572 | 15 | 0.02 | 0.04 | 7.6 | 0.01 | 0.04 | 6.6 |
| TMS | 22.4106 | 114.1244 | 955 | 11 | - | - | - | 0.03 | 0.12 | 10.4 |
| TW | 22.3756 | 114.1267 | 35 | - | - | - | - | 0.11 | 0.22 | 15.7 |
| TWN | 22.3836 | 114.1078 | 142 | - | - | - | - | 0.04 | 0.08 | 16.0 |
| TY1 | 22.3442 | 114.11 | 8 | - | 0.14 | 0.09 | 76.1 | 0.26 | 0.26 | 59.7 |
| TYW | 22.4028 | 114.3231 | 5 | 18 | - | - | - | 0.01 | 0.04 | 7.3 |
| VP1 | 22.2642 | 114.155 | 406 | - | 0.21 | 0.17 | 8.2 | 0.30 | 0.16 | 17.0 |
| WGL | 22.1822 | 114.3033 | 56 | 27 | - | - | - | 0.02 | 0.01 | 6.8 |
| WTS | 22.3394 | 114.2053 | 21 | - | 0.07 | 0.22 | 5.9 | 0.09 | 0.26 | 11.1 |





**Figure B1.** Model evaluation measures in D5 and HW2018.



**Figure B2.** Model evaluation measures in D5 and HW2009.





*Author contributions.*  Robert Schoetter did the main part of the model development, executed the numerical simulations and wrote parts of the Introduction, Sections 2, 3.2, 3.4, 4.1, 4.2, Sections 5.2 to 5.5, and the Conclusions. Yu Ting Kwok wrote the parts of the Introduction related to past mesoscale urban climate model applications to Hong Kong, Sections 3.1, 3.3, 4.3, and 5.1. She also contributed to the testing of the NEW coupling approach. Cécile de Munck contributed to the design of the numerical model setup and the evaluation against the station observations. Kevin Ka Lun Lau contributed to the characterisation of the meteorological station environments. Wai Kin Wong provided

guidance on the HKO observation data. Valéry Masson provided guidance on the model development. All authors read the entire manuscript.

*Competing interests.*  The authors declare no competing interest.

*Acknowledgements.*  Jeanine Payart is acknowledged for the preparation of the ECMWF high-resolution operational forecast analyses. Cindy Lebaupin Brossier is acknowledged for her advice on the SST analysis data. The Hong Kong Observatory is acknowledged for providing the meteorological station and the radiosounding observations. This work has received financial support from the Partenariat Hubert Curien

Programme PROCORE-2019 for scientific exchange between France and Hong Kong for the project "The effects of urban development strategies on the urban climate of Hong Kong: An analysis based on numerical modelling" with the references 42552SL and F-CUHK403/18. It was also supported by a grant from the Research Grant Council of the Hong Kong Special Administrative Region, China (Project No. CUR4046-18F). Yu Ting Kwok received funding from the Hong Kong PhD Fellowship Scheme by the Hong Kong Research Grants Council.





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
