# Peer review of "Multi-layer coupling between SURFEX-TEB-v9.0 and Meso-NH-v5.3 for modelling the urban climate of high-rise cities"

_Geoscientific Model Development, 2020_

## Short Comment (SC1) · 8 Jul 2020

Dear authors,

in my role as Executive editor of GMD, I would like to bring to your attention our Editorial version 1.2:

https://www.geosci-model-dev.net/12/2215/2019/

This highlights some requirements of papers published in GMD, which is also available on the GMD website in the 'Manuscript Types' section:

http://www.geoscientific-model-development.net/submission/manuscript_types.html

[Figure]

In particular, please note that for your paper, the following requirements have not been fully met in the Discussions paper:

- "Code must be published on a persistent public archive with a unique identifier for the exact model version described in the paper or uploaded to the supplement, unless this is impossible for reasons beyond the control of authors. All papers must include a section, at the end of the paper, entitled "Code availability". Here, either instructions for obtaining the code, or the reasons why the code is not available should be clearly stated. It is preferred for the code to be uploaded as a supplement or to be made available at a data repository with an associated DOI (digital object identifier) for the exact model version described in the paper. Alternatively, for established models, there may be an existing means of accessing the code through a particular system. In this case, there must exist a means of permanently accessing the precise model version described in the paper. In some cases, authors may prefer to put models on their own website, or to act as a point of contact for obtaining the code. Given the impermanence of websites and email addresses, this is not encouraged, and authors should consider improving the availability with a more permanent arrangement. Making code available through personal websites or via email contact to the authors is not sufficient. After the paper is accepted the model archive should be updated to include a link to the GMD paper."

Sofar the reader only has access to the modified code. Please provide the information, how a potential user gets access to the regular model code versions of SURFEX-TEB and Meso-NH.

Yours,

Astrid Kerkweg

---

## Author Comment (AC1) · 9 Jul 2020

Dear Astrid Kerkweg,

Thanks for your comment. We have now created a software archive on zenodo with the modified source code, the simulation directories, and the postprocessing scripts (https://zenodo.org/record/3937222#.XwcI6qZ5uV4) and added the information on how to get the regular model code versions of SURFEX-TEB and Meso-NH.

Best regards,

Robert Schoetter

---

## Referee Comment (RC1) · Anonymous Referee #1 · 29 Jul 2020

General comments:

The paper 'Multi-layer coupling between SURFEX-TEB-V9.0 and Meso-NH-v5.3 for modelling the urban climate of high-rise cities' describes the implementation of an updated multi-layer SURFEX-TEB land-surface scheme into the Meso-NH model. The multi-layer SURFEX-TEB is evaluated against in-situ observations in the city of Hong Kong and compared to the previous single-layer version of the SURFEX-TEB. The paper highlights the importance of accounting for building drag and horizontal advection within the urban canopy for the correct estimation of temperature, humidity and wind speed in urban areas. Overall, the quality of the paper is good and it investigates a

very important topic. The current model development, presented in this paper, will be a step towards better weather prediction in urban areas. Yet, I still have a series of minor comments/questions (see below).

Specific comments:

Section 2

1.Lines 114-115. Have the authors tested the effect of vertical discretization of the wall surfaces on the temperature and wind within urban canopy in the multi-layer SURFEX-TEB? I can imagine that using a vertical discretization for the wall facet will allow for the calculation of wall heat fluxes that vary with height within the canopy. This might be particularly useful for reproducing accurate atmospheric stability conditions and vertical mixing in urban canyons. Would the benefits of implementing the vertical discretization outweigh the additional computational costs?

2.Lines 168-169. The roughness length for the roof in this study (0.15m) is larger than that used in different urban surface schemes (i.e. WRF-BEP use 0.01m by default). Is there any particular reason for using the 0.15m roughness length and does this have any implication for the exchange of heat and momentum above the roof? Is a similar roughness length (0.15m) used for road surfaces as well?

3. Is the anthropogenic heat flux deposited at the first atmospheric model level or is a different approach used (i.e. uniformly distributed in the canyon etc.)?

Section 3

4. Lines 267-270 The authors decided to use two heatwave periods (1 to 8 September 2009 and 17 to 31 May 2018) to evaluate the performance of the multi-layer SURFEX-TEB scheme. The selection of a heatwave period is certainly justified, as accurate model performance during heat waves is crucial for the estimation of heat stress. However, since the new scheme is to be employed for weather prediction it is essential to know whether the multi-layer scheme (NEW) offers an improvement over the singlelayer scheme (CLASSICAL) during different atmospheric conditions (i.e. rainy, cloudy days) and seasons (i.e. winter). Have the authors compared the performance of the NEW and CLASSICAL model setups under different atmospheric conditions?

5.Lines 317-320 How many of these measurement stations are located within urban canyons? Is there any relation between the location of the measurement stations and the model bias in temperature, wind speed and relative humidity?

Section 4

6. Have the authors tested the differences in the modeled surface energy balance and turbulent heat fluxes between the 3 model setups (CLASSICAL, NEW and SURFFLUX) at any of the measurement stations?

7. Lines 384-386. Why is the in-depth evaluation of the model performance in the KP and HKP measurement stations done for D4, when both stations are located also within D5? I understand that for a consistent bias comparison between all measurement stations (section 4.1.2) D4 is used, as it contains all of them. Yet since KP and HKP are located within D5, I would expect their evaluation to be done at the highest resolution domain. Have the authors tested the model performance at the KP and HKP stations in both D4 and D5? If so, does the analysis in section 4.1.1 lead to similar conclusions if it is done for D5 instead of D4?

8. Lines 393-395. Have the authors verified the use of 0.1 AOD (i.e. using aeronet stations or satellite AOD products) during both periods? During the 2018HW the assumption of 0.1 AOD seems reasonable, but during the 2009HW period there seems to be substantial difference between the observed and modeled incoming shortwave radiation, especially during the later days of the 2009HW (Figure 5).

9. Lines 515-516. The SBL scheme in the CLASSICAL model setup seems to produce extremely high temperature near the surface during noon (14 local time, Figure 11). Considering also the very low wind speed within the canyon, there seems to be insufficient mixing near the surface in the SBL scheme. What mixing length does the SBL scheme use to calculate temperature and wind speed within the urban canopy? Does this have an effect on the vertical mixing? Have the authors tested whether a modification in the mixing length leads to better results for the temperature, wind speed and relative humidity in the CLASSICAL model setup?

Technical corrections:

10. Lines 448-449 The definition of acceptable quality regarding the rmse error for temperature and relative humidity is rather arbitrary and no measure of acceptable quality is proposed for wind speed. I would suggest that the authors remove/replace the terms "acceptable/unacceptable" as they do not add anything significant to the model evaluation. The rmse values are enough to show the improvement in model performance for the new multi-layer scheme.

---

## Referee Comment (RC2) · Anonymous Referee #2 · 10 Aug 2020

The manuscript by Schoetter et al. presented a recent development of SURFEX-TEB for coupling with Meso-NH by introducing a multi-layer approach and evaluated the performance of the coupled system in Hong Kong. The manuscript is well written and easy to follow, which thus merits to be published after a minor revision.

The only moderate concern is the lack of comparison in surface energy fluxes between the new and classical schemes. Besides, a few details need to be clarified before publication, which can be found as follows:

- Section 2.3: Please discuss the implication of uniform wall surface temperature with respect to uncertainties.
- L295: please add Kwok et al. (2020) to the reference list.

- L339: Clarify if this all goes into sensible heating.

- L357: Please clarify how zero-plane displacement is calculated as well.

- L359: Are the other four levels evenly spaced?

- L378: Please explain why this won't be viable when using larger time steps.

- L606: Please provide the prognostic equation, which can go into appendix.

Other technical comments:

- Figure 1: Better to align the basement levels of two approaches at the level: it is understandable that the new approach would apply for grids with much higher buildings.

- Equation 3: Correct the less than or equal sign to "$\leq$" here and other occurrences.

- L181: please use scientific notation for the numbers.

- L185: "explicited" -> "explicitly given"

- Figure 4: Use dots to represent observations for better contrast and easier legibility.

- Figure 13: correct the unit in y label to be consistent with main text.
* * *

---

## Author Comment (AC2) · 4 Sep 2020

**Response to the reviewers comments for the manuscript "Multi-layer coupling between SURFEX-TEB-V9.0 and Meso-NH-v5.3 for modelling the urban climate of high-rise cities"**

**Referee #1**

General comments:
The paper 'Multi-layer coupling between SURFEX-TEB-V9.0 and Meso-NH-v5.3 for modelling the urban climate of high-rise cities' describes the implementation of an updated multi-layer SURFEX-TEB land-surface scheme into the Meso-NH model. The multi-layer SURFEX-TEB is evaluated against in-situ observations in the city of Hong Kong and compared to the previous single-layer version of the SURFEX-TEB. The paper highlights the importance of accounting for building drag and horizontal advection within the urban canopy for the correct estimation of temperature, humidity and wind speed in urban areas. Overall, the quality of the paper is good and it investigates a very important topic. The current model development, presented in this paper, will be a step towards better weather prediction in urban areas. Yet, I still have a series of minor comments/questions (see below).
Thank you for taking the time to provide such a careful review of our submission. Following your suggestions, we have investigated the surface energy balance (SEB) at two of the stations (King's Park and Hong Kong Park) and found only small differences for the different coupling approaches. The large differences in the near-surface meteorological variables for the different approaches are therefore due to the way SURFEX-TEB is coupled to Meso-NH and not due to differences in the SEB. Furthermore, we now provide more information on the urban turbulent mixing length and we also conducted an additional simulation to test a different formulation of the mixing length. Results from this additional simulation are however not that different from the CLASSICAL simulation presented in detail in the manuscript. This finding, together with the other simulations and budget analysis indicates that the largest issue with the single-layer coupling is the lack of horizontal advection in the urban canopy layer. We give the detailed answers to your comments below and provide a version of the manuscript with highlighted modifications.

Specific comments:
Section 2

Lines 114-115. Have the authors tested the effect of vertical discretization of the wall surfaces on the temperature and wind within urban canopy in the multi-layer SURFEX-TEB? I can imagine that using a vertical discretization for the wall facet will allow for the calculation of wall heat fluxes that vary with height within the canopy. This might be particularly useful for reproducing accurate atmospheric stability conditions and vertical mixing in urban canyons. Would the benefits of implementing the vertical discretization outweigh the additional computational costs?
We cannot test the vertical discretization of the wall surfaces since this would require major additional model developments. In the case, the wall temperature is vertically discrete, the

calculation of radiative exchange between the walls and other facets (opposite wall, impervious, vegetation) becomes far more complex than in the current version. These developments are out of the scope of the present study. Along with the comment of Reviewer #2, we have included a point on the neglect of the vertical discretization of the wall in Section 2.3.

Lines 168-169. The roughness length for the roof in this study (0.15 m) is larger than that used in different urban surface schemes (i.e. WRF-BEP use 0.01 m by default). Is there any particular reason for using the 0.15 m roughness length and does this have any implication for the exchange of heat and momentum above the roof? Is a similar roughness length (0.15 m) used for road surfaces as well?
The aerodynamic roughness length for the roofs of 0.15 m shall not only represent the flat tiles/concrete on the roof for which 0.01 m would be justified, but also potential chimneys, air conditioning systems, and other small constructions that are usually present on the roofs. The value of the aerodynamic roughness length for the roads is 0.05 m. We now point this out in the manuscript.

Is the anthropogenic heat flux deposited at the first atmospheric model level or is a different approach used (i.e. uniformly distributed in the canyon etc.)?
We did not describe with sufficient detail how the anthropogenic heat flux is injected in the model. It depends on the source of the anthropogenic heat flux. The heat flux due to traffic and industrial activities is injected at the first atmospheric model level for both single- and multi-layer coupling. The anthropogenic heat flux due to building heating, electrical appliances, cooking, lighting is injected inside the building. It reaches the atmosphere indirectly in two ways: 1) heat conduction through the building envelope and subsequent infrared radiation and turbulent sensible heat exchange between the building facets (walls, roofs, windows) and the atmosphere as well as 2) air exchange due to infiltration, natural and mechanical ventilation. For buildings with a heating system based on combustion inside the building (e.g. gas, fuel, or wood burning), the waste heat and moisture fluxes are directly injected into the atmosphere at roof level (chimneys). The roof level is the SBL level (atmospheric level) intersecting the roof for the single-layer (multi-layer) coupling. The waste heat and moisture fluxes due to air conditioning can be injected at wall level for wall split air conditioning systems or roof level for cooling tower based air conditioning systems. The wall level fluxes are distributed evenly over the SBL levels (atmospheric model levels) intersecting the walls for the single-layer (multi-layer) coupling. The roof level fluxes are injected at roof level. These details are now provided in the manuscript (Section 3.2).

Section 3
Lines 267-270: The authors decided to use two heatwave periods (1 to 8 September 2009 and 17 to 31 May 2018) to evaluate the performance of the multi-layer SURFEX-TEB scheme. The selection of a heatwave period is certainly justified, as accurate model performance during heat waves is crucial for the estimation of heat stress. However, since the new scheme is to be employed for weather prediction it is essential to know whether the multi-layer scheme (NEW) offers an improvement over the single-layer scheme (CLASSICAL) during

different atmospheric conditions (i.e. rainy, cloudy days) and seasons (i.e. winter). Have the authors compared the performance of the NEW and CLASSICAL model setups under different atmospheric conditions?

No, currently the multi-layer coupling has not been tested for various meteorological situations, different seasons, and a variety of cities. This is planned in subsequent studies but out of the scope of the present study. We now mention the need for further testing of the multi-layer coupling in the conclusions and outlook section. We furthermore speculate that the benefit from the multi-layer coupling will be lower for meteorological situations with higher wind speed and cloudy conditions, since for such situations the urban heat island intensity is lower than for situations with clear sky and low wind speed. Furthermore, the difference between the single- and multi-layer coupling can also be expected to be smaller for low to mid-rise cities than for the high-rise city of Hong Kong.

Lines 317-320 How many of these measurement stations are located within urban canyons? Is there any relation between the location of the measurement stations and the model bias in temperature, wind speed and relative humidity?

None of the measurement stations is directly located in an urban canyon. This is an important point since studies based on obstacle-resolving models developing formulations, e.g. for the urban turbulent mixing length scale are made for areas representative of the space in between the buildings. For a fully rigorous evaluation of such formulations in real cities, also observations representative of such areas would be needed, which is rarely the case. We already point this out in the discussion (Section 5.4) and in the penultimate paragraph of the conclusion and outlook section. Concerning the relation between the model performance and the location of the stations, the most relevant outcomes are discussed in Section 4.1.2. Model performance for the single-layer coupling is worst for those stations surrounded by high-rise buildings and the stations located in a very heterogeneous urban environment. For these stations, the model improvement for the multi-layer coupling is also largest. For the SURFFLUX coupling approach (multi-layer coupling, but heat and moisture fluxes released at the surface), the model performance is only deteriorated compared to the NEW coupling approach for stations in the vicinity of buildings higher than 40 m.

Section 4
Have the authors tested the differences in the modeled surface energy balance and turbulent heat fluxes between the 3 model setups (CLASSICAL, NEW and SURFFLUX) at any of the measurement stations?

In the new Section 5.3, we now analyse the simulated surface energy balance (SEB) for the 3 coupling approaches (CLASSICAL, NEW and SURFFLUX) in a 1 km x 1 km box centred on the stations KP and HKP. The SEB does not differ in a relevant way between the different coupling approaches. This shows that the large differences in the near-surface meteorological variables between the NEW (multi-layer) and CLASSICAL (single-layer) coupling approach are not due to changes in the SEB, but due to the different way the surface fluxes are coupled with the atmospheric model.

Lines 384-386. Why is the in-depth evaluation of the model performance in the KP and HKP measurement stations done for D4, when both stations are located also within D5? I understand that for a consistent bias comparison between all measurement stations (section 4.1.2) D4 is used, as it contains all of them. Yet since KP and HKP are located within D5, I would expect their evaluation to be done at the highest resolution domain. Have the authors tested the model performance at the KP and HKP stations in both D4 and D5? If so, does the analysis in section 4.1.1 lead to similar conclusions if it is done for D5 instead of D4?

The in-depth evaluation in Section 4.1.1 is indeed done for D4 such that the model performance can be directly compared with the summary plots with all stations in D4 in Section 4.1.2. The summary plots for D5 are given in the Appendix Figures B1 and B2. Overall, the results for D5 are very similar to those for D4. For single stations and meteorological variables they can be a bit different, which is probably due to the changing representativeness of the model grid cell between 250 m x 250 m and 125 m x 125 m. We mentioned the Figures B1 and B2 at the beginning of Section 4.1.2, but we missed to describe them, which we now do briefly at the end of this section. The detailed analysis in Section 4.1.1 for the stations KP and HKP would lead to similar conclusions if it would be done for D5.

Lines 393-395. Have the authors verified the use of 0.1 AOD (i.e. using aeronet stations or satellite AOD products) during both periods? During the 2018HW the assumption of 0.1 AOD seems reasonable, but during the 2009HW period there seems to be substantial difference between the observed and modeled incoming shortwave radiation, especially during the later days of the 2009HW (Figure 5).

Based on the reviewers suggestion, we investigate the AOD from TERRA/MODIS (https://neo.sci.gsfc.nasa.gov/view.php?datasetId=MODAL2_M_AER_OD).
It is generally challenging to determine representative values for the area of Hong Kong since it lies in transition zone between typically very high AOD values over continental China and much cleaner air over the South China Sea. During the two selected heat waves, the mesoscale wind direction is east and south-west for HW2009 and HW2018 respectively, with therefore advection from the South China Sea.
For HW2009, the TERRA/MODIS AOD values over the sea close to Hong Kong lie in between 0.2 and 0.4, thus larger than the value used in the model. They can reach up to 1.0 over continental China. For HW2018, the AOD values lie between 0.0 to 0.2 over the sea close to Hong Hong. Apart from potential local aerosol emissions, the air was therefore quite clean during HW2018.
The analysis of the AOD values from TERRA/MODIS thus confirms the reviewer's suspicion that AOD values of 0.1 used in the simulations are good for HW2018, but too low for HW2009. We therefore modulate the discussion in Section 4.1.1.

Lines 515-516. The SBL scheme in the CLASSICAL model setup seems to produce extremely high temperature near the surface during noon (14 local time, Figure 11). Considering also the very low wind speed within the canyon, there seems to be insufficient mixing near the surface in the SBL scheme. What mixing length does the SBL scheme use to calculate temperature and wind speed within the urban canopy? Does this have an effect on

the vertical mixing? Have the authors tested whether a modification in the mixing length leads to better results for the temperature, wind speed and relative humidity in the CLASSICAL model setup?

We agree that more attention needs to be paid to the turbulent mixing length in the urban SBL scheme. For the CLASSICAL experiment described in the present manuscript, the urban mixing length is calculated following Santiago and Martilli (2010). We now provide this information in the description of the CLASSICAL experiment (Section 3.4). Furthermore, we conducted an additional simulation modifying the mixing length in the urban SBL scheme to be equal to the distance from the surface, as it is in the atmospheric model. The results, which are very similar to those obtained for the CLASSICAL experiment are briefly discussed in Section 5.4.

Technical corrections:

Lines 448-449 The definition of acceptable quality regarding the rmse error for temperature and relative humidity is rather arbitrary and no measure of acceptable quality is proposed for wind speed. I would suggest that the authors remove/replace the terms "acceptable/unacceptable" as they do not add anything significant to the model evaluation. The rmse values are enough to show the improvement in model performance for the new multi-layer scheme.

Done.

---

## Author Comment (AC3) · 4 Sep 2020

**Response to the reviewers comments for the manuscript "Multi-layer coupling between SURFEX-TEB-V9.0 and Meso-NH-v5.3 for modelling the urban climate of high-rise cities"**

**Referee #2**

The manuscript by Schoetter et al. presented a recent development of SURFEX-TEB for coupling with Meso-NH by introducing a multi-layer approach and evaluated the performance of the coupled system in Hong Kong. The manuscript is well written and easy to follow, which thus merits to be published after a minor revision. The only moderate concern is the lack of comparison in surface energy fluxes between the new and classical schemes. Besides, a few details need to be clarified before publication, which can be found as follows: Thank you for taking the time to provide such a careful review of our submission. We have added the comparison of the surface energy fluxes at the stations KP and HKP for the different coupling approaches (new Section 5.3) and conducted a simulation with the single-layer coupling and another formulation for the turbulent mixing length in the SBL scheme as suggested by Reviewer #1. We give the detailed answers to your comments below and provide a version of the manuscript with highlighted modifications.

- Section 2.3: Please discuss the implication of uniform wall surface temperature with respect to uncertainties.
Done.

- L295: please add Kwok et al. (2020) to the reference list.
This article is now published online and we add the correct reference.

- L339: Clarify if this all goes into sensible heating.
In Section 3.3.2, we describe how we spatially disaggregate the monthly energy consumption to calculate the inventory-based anthropogenic heat flux. The values calculated in this Section are not directly used in the model since it calculates the building energy consumption for air conditioning based on the prevailing meteorological conditions, the characteristics of the building envelope, and the setpoint temperature of air conditioning given in Kwok et al. (2020). The anthropogenic heat flux calculated by the model contains a sensible and a latent part. For the internal heat release it is considered that a part of it is latent (e.g. cooking, domestic hot water). The latent fraction of the internal heat release is specified as a function of the building type and is 0.05 for schools, 0.1 for university buildings, 0.2 for shopping malls, industrial buildings, and office buildings, and 0.3 for residential and public health buildings. For the energy consumption due to air conditioning, it is considered that there might be evaporative cooling towers on the building roofs. The fraction of buildings equipped with such cooling towers is specified as a function of the building type and is 0 for most buildings, except for schools and other Government, Institutional, and Community buildings (0.1), commercial and public health buildings (0.2), and office buildings (0.5). We now provide this information in Section 3.2.

- L357: Please clarify how zero-plane displacement is calculated as well.
We now provide the equation for the zero-plane displacement height (Equation 25) together with the more detailed description of the urban turbulent mixing length in Section 3.4.

- L359: Are the other four levels evenly spaced?
No, the depth of the SBL levels increases with height. This is now stated in the manuscript.

- L378: Please explain why this won't be viable when using larger time steps.
The reason is that the temperature and moisture increments would become too larger. We now mention this in the manuscript.

- L606: Please provide the prognostic equation, which can go into appendix.
We now provide the prognostic equation for potential temperature in the appendix.

Other technical comments:
- Figure 1: Better to align the basement levels of two approaches at the level: it is understandable that the new approach would apply for grids with much higher buildings.
We prefer to not align the basement levels of the two approaches since we want to highlight that, with the single-layer coupling, the urban canopy layer is located below the physical and atmospheric model surface. We change the figure and the caption slightly to make this more clear.

- Equation 3: Correct the less than or equal sign to "$\leq$" here and other occurrences.
The proper Latex \leq command is now used in all equations.

- L181: please use scientific notation for the numbers.
Done.

- L185: "explicited" -> "explicitly given"
Done.

- Figure 4: Use dots to represent observations for better contrast and easier legibility.
We now use dots to represent the observations in all time series plots.

- Figure 13: correct the unit in y label to be consistent with main text.
Done.

---

## Author Response (AR2)

**Response to the topical editor's comment for the manuscript "Multi-layer coupling between SURFEX-TEB-v9.0 and Meso-NH-v5.3 for modelling the urban climate of high-rise cities"**

Dear Authors,
Thank you for your comprehensive answers to the referee comments. I'm ready to accept the paper for final publications after you have updated the Code and Data availability section to match with the dataset in Zenodo you have created. This has the added text requested by the executive editor. This update is not included in the document in the supplementary. I had a look on the description and concerning the high-resolution operational data it is said that it is available upon request. Are these original data freely available from ECMWF? If yes, you could link the dataset to the document. In that case the user should be able to download the data as instructed, and run PREP_REAL_CASE program to produce appropriate lateral forcing and this should be enough.

Dear Leena Järvi,

Thank you for taking the time to edit our revised manuscript. We indeed missed to refer to the newly created zenodo archive in the revised version of our manuscript and we do this now in the code and data availability section. Since zenodo allows to attach up to 50 Gb per dataset, we now also provide the ECMWF data for a domain covering southern China and the South China Sea. We do no longer include a Supplement, since it is entirely replaced by the zenodo archive.

Best regards,

Robert Schoetter on behalf of the authors

[revised manuscript text omitted]